METHODS AND RESOURCES

# Cell type-agnostic transcriptomic signatures enable uniform comparisons of neural maturation

Sridevi Venkatesan[1,2,3], Jonathan M. Werner[3], Yun Li[2,4], Jesse Gillis[1,3,4]*

1 Department of Physiology, University of Toronto, Toronto, Canada, 2 Developmental and Stem Cell Biology, Hospital for Sick Children, Toronto, Canada, 3 Terrence Donnelly Centre for Cellular and Biomolecular Research, University of Toronto, Toronto, Canada, 4 Department of Molecular Genetics, University of Toronto, Toronto, Canada

* jesse.gillis@utoronto.ca

## Abstract

Understanding where a cell sits along developmental time is as important as identifying its type. While single-cell transcriptomics has catalogued the diversity of neural cell types, aligning them along a shared temporal axis across studies, species, and model systems remains a fundamental challenge. Here, we develop a single-cell transcriptomic 'clock' that predicts true developmental age, enabling standardized, cross-context comparisons of neural maturation. Through a meta-analysis of over 2.8 million cells from the developing human brain, we identify robust tissue-level and cell-autonomous predictors of developmental age. We find that bulk tissue composition predicts age within individual studies but lacks generalizability, whereas specific cell type proportions, particularly astrocytes and progenitors, track age reliably across studies. Using machine learning, we develop a cell type-agnostic predictor based on 462 genes that robustly tracks developmental dynamics across diverse cell types and datasets (error = 2.6 weeks). Our model accurately estimates developmental age in human neural organoids and detects disease-associated shifts. Model predictions further generalize across species, revealing 10-fold accelerated neurodevelopment in mice relative to humans. Our approach provides a robust framework to assess neural maturation across contexts, with broad relevance for developmental biology and disease modeling.

## Introduction

The developing mammalian brain still lacks a quantitative scale that can be applied universally to gauge how far any given cell has progressed along its maturational trajectory. In other domains, conceptual breakthroughs have followed the invention of such scales. Akin to a thermometer for temperature, biological aging was transformed by the Horvath epigenetic clock, which converts complex methylation states into a single "biological-age" number [1–3]. By contrast, studies of neurodevelopment continue to rely on disparate and incompatible proxies: gestational week in fetal tissue,

**Data availability statement:** This paper analyzes existing, publicly available data, accessible at links provided in S1 Table. All original code and data used to generate figures are available at https://github.com/sridevi96/NeuroDevTime and https://doi.org/10.5281/zenodo.14908185.

**Funding:** This work was supported by the National Institutes of Health (https://www.nih.gov/, U24MH130968 to JG and JMW; R01MH133181 to JG; R01MH113005 to JG), the Canadian Institutes of Health Research (https://cihr-irsc.gc.ca/e/193.html, PJT-180565 to YL), and Schmidt Science Fellows (https://schmidtsciencefellows.org/, postdoctoral fellowship to SV). The funders had no role in study design, data collection and analysis, decision to publish, or preparation of the manuscript.

**Competing interests:** The authors have declared that no competing interests exist.

**Abbreviations:** ASD, autism spectrum disorder; FDR, false discovery rate; GO, Gene Ontology; HNOCA, Human Neural Organoid Cell Atlas; MAE, Mean Absolute Error; PCs, Principal components.

days in vitro for organoids, embryonic day in model organisms, or ad-hoc pseudotime axes computed separately within each single-cell data set [4–9]. The absence of a standardized ruler hampers direct comparison across protocols, laboratories, brain regions, species, and disease models.

Existing transcriptomic approaches illustrate the problem. Principal-component or pseudotime analyses capture orderly gene-expression waves inside one experiment [10–12], but collapse when confronted with different cell-type mixtures, sequencing platforms, or anatomical regions [13,14]. Because there is no universal unit of pseudotime, aligning developmental timing across such contexts requires batch correction or time-warping steps that risk removing biological information [15]. Sequential differential-expression methods and label-transfer tools likewise generate study-specific developmental programs whose axes are not commensurable across data sets [6,16–19]. A central reason is that single-cell profiles encode two intertwined signals: (i) large-scale shifts in tissue composition as progenitors give way to neurons and glia, and (ii) cell-intrinsic maturation programs that unfold within each lineage.

Disentangling these signals and learning which components generalize across studies has remained an open challenge. To address this gap, we performed a comprehensive meta-analysis of 13 publicly available single-cell RNA-sequencing data sets that together comprise more than 2.8 million cells from the first to third trimesters of human brain development. These datasets span multiple brain regions, including the cortex, ganglionic eminences, and hypothalamus [6,20–31]. Using cross-validated machine-learning models, we quantified two orthogonal predictors of developmental age. At the tissue-level, we show that a simple compositional index—principally the opposing trajectories of astrocytes and neural progenitors—tracks chronological age across laboratories and sequencing platforms, whereas more elaborate composition-based models fail to generalize. At the cellular level, a regularized regression model trained on a compact set of ~ 460 genes captures a conserved, cell-type-agnostic maturation axis that predicts human gestational week with an error of 2–3 weeks, independent of brain region, cell class, or technology. Because the model is agnostic to cell-type labels and requires no batch integration, it can be applied out of the box to any new single-cell data set.

Our transcriptomic clock yields immediate biological insights. Applied to cortical organoids, it reveals accelerated maturation of GABAergic progenitors in autism-linked genetic backgrounds and quantifies the developmental boost conferred by transplantation into rat cortex. When restricted to one-to-one orthologues, the model recapitulates the well-known but previously hard-to-measure acceleration of neurodevelopment in mice relative to humans, aligning embryonic day 18 mouse neurons to roughly gestational week 21 in human. More broadly, the clock furnishes a common temporal axis against which protocol fidelity, disease-associated timing shifts, and cross-species tempo differences can be measured quantitatively. By providing a standardized, generalizable ruler for cellular age, our work aims to place neurodevelopmental studies on the same footing that epigenetic clocks afforded to

aging research, enabling rigorous comparison across the expanding landscape of single-cell atlases and stem-cell-derived model systems.

## Results

We perform a meta-analysis of 13 recent single-cell RNA-seq datasets from the developing human brain, spanning 151 donors across the first to third trimesters, and comprising 2,888,635 cells (Fig 1, detailed dataset information in S1 Table). We evaluate the relative contributions of tissue-level cellular composition and cell-autonomous transcriptomic maturation in predicting developmental age across diverse contexts.

### Tissue-level composition predicts neurodevelopmental age

**Study-specific developmental changes in cell type composition do not generalize.** Neurodevelopment is accompanied by large-scale shifts in cell type composition, following the well-established sequence of progenitor proliferation, neurogenesis, and subsequent gliogenesis [32–34]. We first evaluated whether tissue-level cell type composition in the human brain reliably predicts developmental age across multiple datasets. To standardize comparisons, we grouped author-provided cell type annotations into seven broad cell classes based on consensus marker genes (Figs 2A and S1 and S2 Table). We further verified that major cell classes were highly replicable across all datasets (AUROCs = 0.8–0.9, S2 Fig) using MetaNeighbor [35]. Regularized regression models were then trained to predict sample age from cell class proportions within each study. Study-specific models performed well within individual datasets, accurately predicting gestational age in 7 of 11 studies, with strong correlations between predicted and actual ages (best performance: Braun, R = 0.94, $P = 2.6e-12$; S3 Fig). Despite high within-study-accuracy (median error = 1.19 weeks), these models failed to generalize across datasets, with prediction error increasing nearly 8-fold when applied to unseen studies (median error = 9.13 weeks, Wilcoxon test, $P = 2.8 \times 10^{-6}$; Fig 2B). Thus, while developmental changes in cell type composition are robust within individual studies, they do not generalize well across datasets.

**Principal component age predictions are primarily driven by cell type composition.** Principal components (PCs) derived from gene expression are commonly used to compare developmental age within a study, with PC1 typically showing strong correlation to age [4,5,9,10]. However, because cell type composition is the dominant source of transcriptomic variation during development, it may underlie PC1-based age predictions. Given the poor generalizability of study-specific compositional changes, we assessed whether PC-based age predictions remain accurate across datasets using PCs obtained from pseudo-bulked transcriptomes.

In all eight studies tested, either PC1 (50%) or PC2 (50%) correlated significantly with age, enabling reliable within-study age prediction (median error = 1.95 weeks, S4 Fig). However, projecting other studies onto the age-correlated PC identified within a single study resulted in significantly worse performance, with a 3-fold increase in error (median error = 6.04 weeks, Wilcoxon test, $P = 0.0003$; Fig 2C). Since cell type composition varies across datasets, we examined the relationship between age-correlated PCs and cell type proportions. Strikingly, the correlation between PCs and age was nearly identical to their correlation with various cell type proportions (e.g., Astrocytes: R = 0.88, $P = 2 \times 10^{-12}$; Progenitor: R = −0.69, $P = 1.1 \times 10^{-6}$; Oligo: R = 0.81, $P = 1.7 \times 10^{-10}$; Immune: R = 0.78, $P = 2.9 \times 10^{-9}$; S4 Fig).

To directly assess how study-specific cell type enrichment or depletion affects composition-based age signals, we performed a simulation: within each study, we removed all cells belonging to a single-cell type, recomputed pseudo-bulk expression, and re-derived the PCs. We then tested whether the age-associated PC remained stable. Strikingly, eliminating a single-cell type was sufficient to alter or abolish the age-correlated PC in **4 out of 8 studies** (S5 Fig). This mirrors our cross-study projection results, where a PC that tracks age in one dataset fails to generalize to another. These results show that modest compositional differences substantially alter PC structure and its relationship to age, making cross-study generalization fragile.

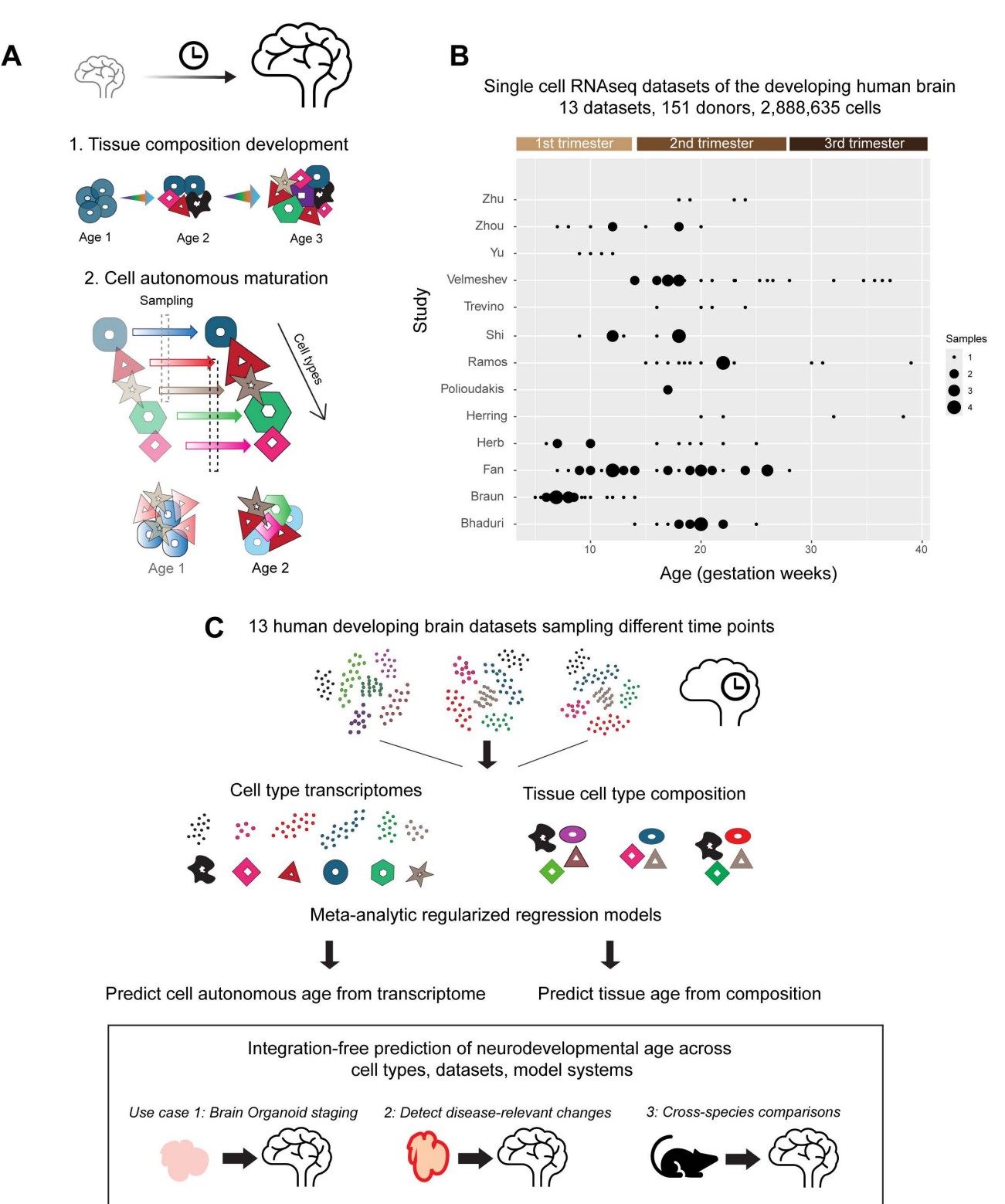

**Fig 1. Meta-analysis of human brain single-cell data to identify tissue-level and cell-autonomous predictors of neurodevelopmental age.**
**A**, Neurodevelopment involves tissue-level changes in brain cell type composition and cell-autonomous transcriptomic maturation within individual cell types. Using a meta-analysis of recent single-cell datasets from the fetal human brain, we independently evaluate these features to identify robust patterns that consistently predict neurodevelopmental age across datasets. **B**, Distribution of sample ages in 13 fetal brain datasets included in

meta-analysis. **C**, Summary of workflow: Meta-analytic regularized regression models are trained to identify cell-autonomous or tissue-level predictors of neurodevelopmental age. We show their application for i) Assessing developmental progression in human neural organoids; ii) Identifying disease-associated shifts in mutant organoids, and iii) Aligning neurodevelopmental trajectories across human and mouse brain development.

**Astrocyte and progenitor proportions consistently predict neurodevelopmental age across datasets.** Rather than relying on global tissue composition, which varies across studies, we focused on well-defined cell types whose proportions are more consistent and reliably predict developmental age. Using feature selection, we identified the best-performing cell classes for age prediction across datasets. Linear models using only progenitors, or a combination of progenitors and astrocytes, achieved the best performance across studies in a leave-study-out cross-validation (Mean error across cross-validation folds, using progenitors: 4.26 weeks, $P=0.014$ compared to null model; using astrocytes and progenitors: 4.14 weeks, $P=0.006$; Fig 2D). The meta-analytic model trained using astrocyte and progenitor cell class proportions reliably predicts age across 9 studies (correlation between predicted and actual age of samples: R=0.68, $P<2.2\times10^{-16}$, Fig 2E). We further tested this pre-trained model on two held-out datasets from the developing human hypothalamus [22,27], achieving accurate predictions of age in both studies (correlation between predicted and actual ages: Herb: R=0.95, $P=5.2\times10^{-6}$; Zhou: R=0.83, $P=0.005$; Fig 2E).

A simple compositional model using astrocyte and progenitor cell class proportions effectively predicts age in held-out datasets with least error (median error=4.52 weeks), compared to study-specific composition (median error=9.13 weeks; Wilcoxon test: $P=0.001$), or PCA (median error=6.03 weeks; Wilcoxon test: $P=0.007$; Fig 2F). This model relies on the increase in astrocyte proportions and the decrease in progenitor proportions during development, depicted with a positive coefficient for astrocytes (35.32) and negative coefficient for progenitors (−22.46).

In sum, study-specific tissue composition predicts neurodevelopmental age within individual studies, but cross-study variability prevents this from generalizing across datasets. In contrast, astrocyte and progenitor proportions exhibit consistent developmental trajectories across datasets, enabling robust and generalizable age predictions. These relative shifts in astrocyte and progenitor abundance are sufficient to predict tissue age across studies.

We next turned our attention to cell-autonomous predictors of developmental age trained on single-cell transcriptomes that are independent of tissue composition.

## Cell type-agnostic transcriptomic signatures enable reliable predictions of cell-autonomous neurodevelopmental stage

Single-cell transcriptomic age predictors have enabled reliable predictions of biological age in aging tissues from mice [36] and humans [37]. We aimed to develop a transcriptomic age predictor for cell types in early human brain development to uniformly compare cell-autonomous maturation across datasets. To mitigate the impact of single-cell data sparsity, we used log-normalized expression from cell type-aggregated transcriptomes to train meta-analytic regularized regression models. Models were first trained on each specific cell type to predict age in held-out datasets (leave-study-out cross-validation, Fig 3A). Cell type-specific models reliably predicted the age of the cell class they were trained on across datasets (Median error across cross-validation folds in weeks: Astrocyte: 4.10, Progenitor: 2.92, Excitatory Neuron: 2.27, Inhibitory neuron: 2.31, Oligodendrocytes: 3.82, Vascular: 5.86, Immune: 4.05, and Other: 4.58; Fig 3B). However, cell type-specific models also showed equivalent age prediction accuracy in cell types that they were not trained on, revealing a surprising lack of cell type-specificity (median error on trained on cell types: 3.94 weeks, median error on other cell types: 3.92 weeks; Wilcoxon test: $P=0.25$, Fig 3C).

Given the lack of cell type-specificity observed above, we deliberately trained a cell type-agnostic model on transcriptomes from all cell classes across datasets to predict age. By equally weighing every cell type within each study, this cell

PLOS Biology

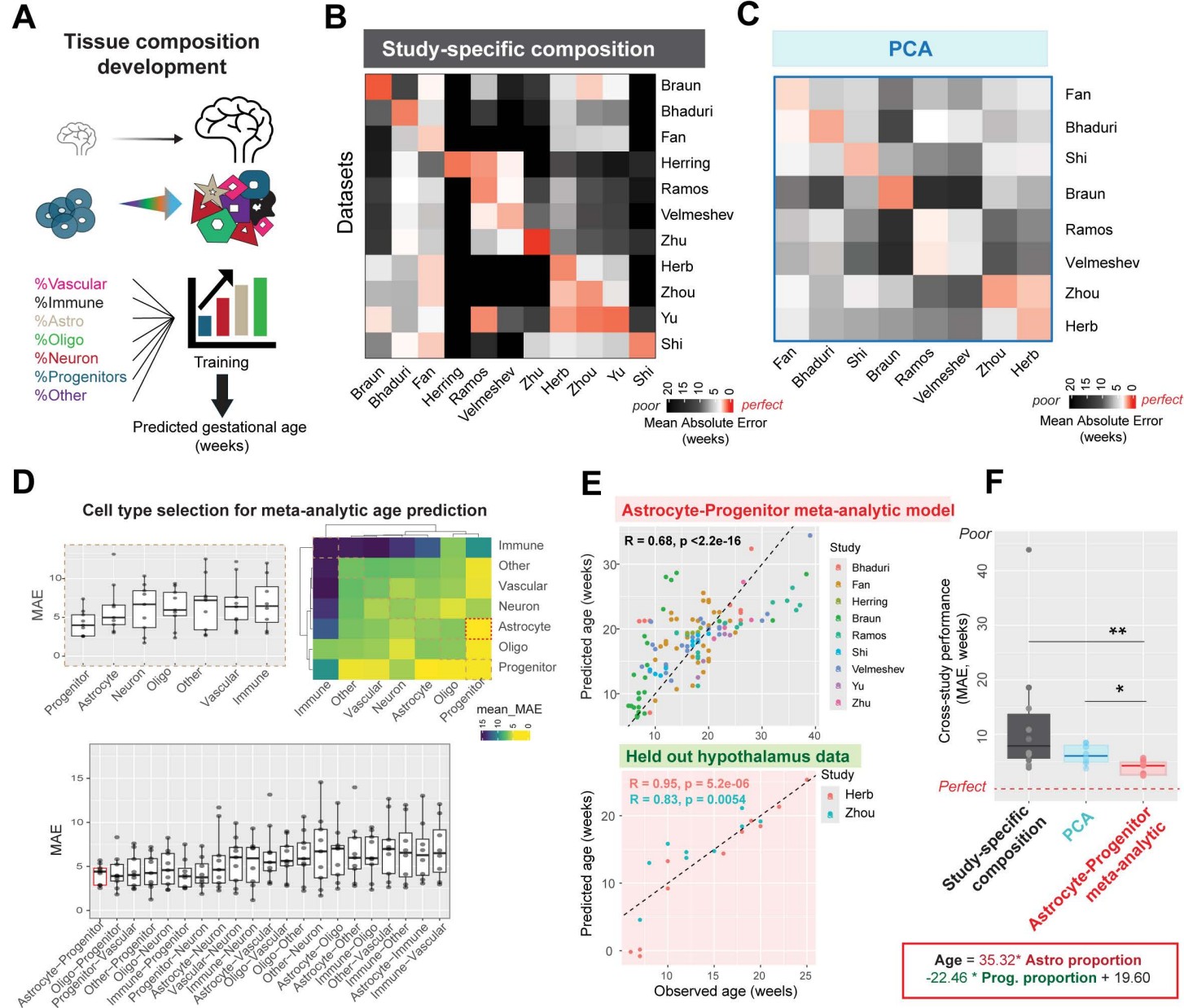

**Fig 2. Tissue-level changes in cell type composition predict neurodevelopmental age, with study-specific differences impacting generalizability. A**, Proportions of 7 broad cell classes- Vascular, Immune, Astrocyte, Oligodendrocyte, Neuron, Progenitor, and Others were used to train regularized regression models to predict sample age from tissue composition within individual studies. **B**, Within-study and cross-study performance (Mean Absolute Error in weeks) of study-specific compositional models (leave-sample-out cross-validation, see S2 Fig). **C**, Performance of models trained on principal components (PCs): Linear models were trained on age-correlated PCs within each study derived from pseudo-bulked transcriptomes (see S3 Fig). The heatmap shows within-study and cross-study performance of PC-based age prediction. **D**, Linear models trained on proportions of specific cell types were evaluated for their ability to predict tissue age meta-analytically across datasets. Progenitors alone (top left) or the combination of progenitors and astrocytes (bottom) show least mean absolute error (MAE). The heatmap (top right) displays individual cell type performance along the diagonal and cell type combination performance off the diagonal. **E**, Performance of meta-analytic model trained on astrocyte and progenitor proportions in leave-study-out cross-validation (top). This model shows good performance in two held-out datasets from the human hypothalamus (bottom). **F**, Comparison of age prediction performance (MAE) across held-out studies for three model types: study-specific composition, PCA, and astrocyte-progenitor proportions. The axis is restricted, with one value (792.63) from study-specific composition performance omitted for clarity. (*$P < 0.05$, **$P < 0.01$, Wilcoxon test).

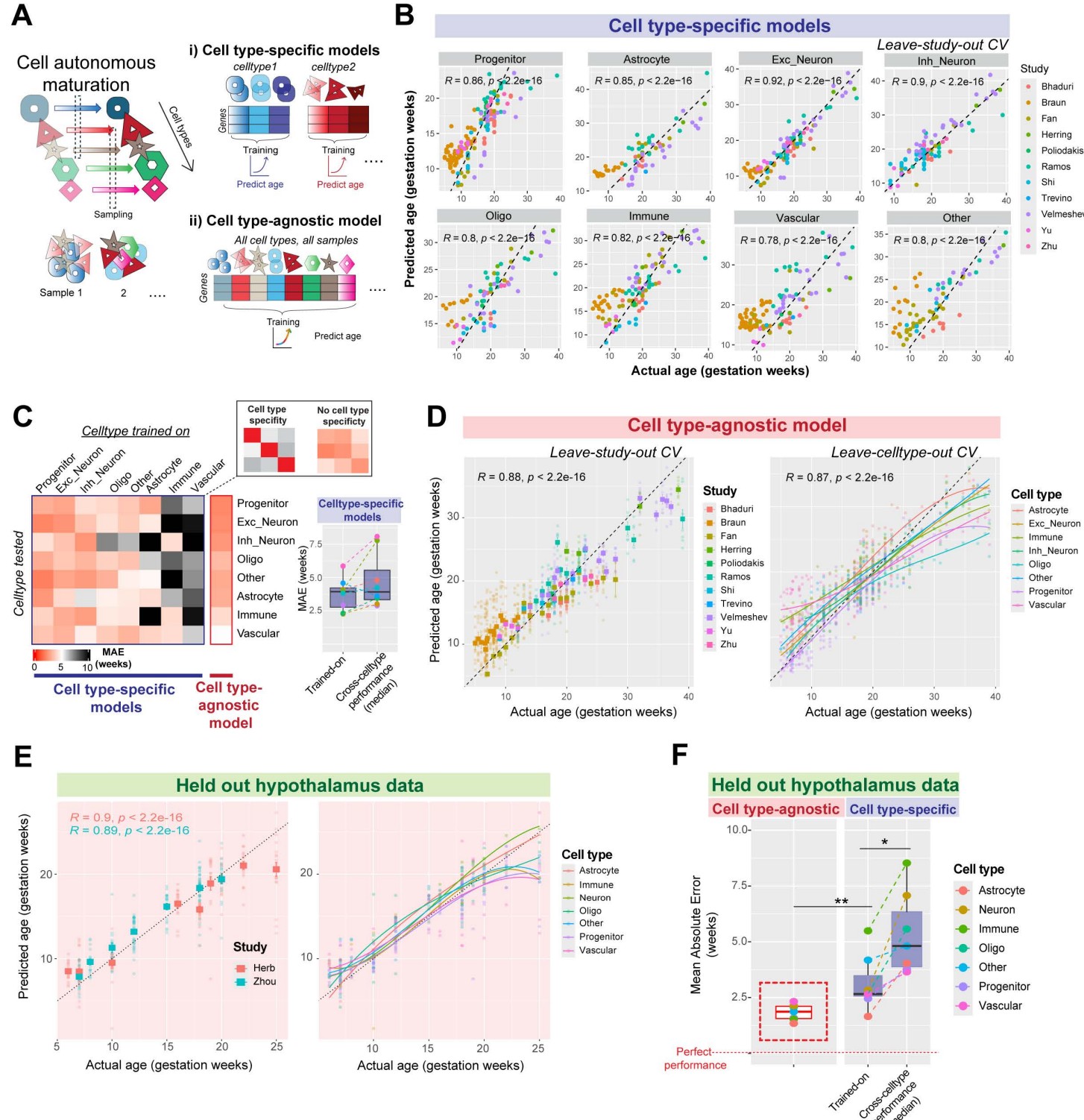

**Fig 3. A cell type-agnostic model robustly predicts neurodevelopmental age across cell types and datasets. A,** Cell-autonomous transcriptomic maturation occurs within individual cell types during neurodevelopment and distinct samples across datasets may capture cells in different states of maturation. We trained meta-analytic cell type-specific models on individual cell types, or a cell type-agnostic model jointly on all cell types to predict age from log-normalized gene expression. **B,** Performance of cell type-specific models in predicting neurodevelopmental age of cell types from held-out studies (leave-study out cross-validation). **C,** Heatmap shows cross-cell type performance of cell type-specific models (left), revealing poor cell

type-specificity, akin to the performance of a cell type-agnostic model. The boxplot (right) compares the performance of each cell type-specific model on its trained cell type with median performance across all other cell types. **D,** Performance of cell type-agnostic model on predicting age across held-out studies (leave-study out cross-validation, left) and across held-out-cell types (leave-cell type out cross-validation, right). **E,** Cell type-agnostic model shows robust performance in two held-out hypothalamus datasets not used in training. **F,** Comparison of mean absolute error (MAE) in age prediction within held-out hypothalamus datasets by the cell type-agnostic model (left) and cell type-specific models (right, within and across cell types; *$P<0.05$, **$P<0.01$, Wilcoxon test).

type-agnostic model learned a general signature of neurodevelopment, showing robust performance in predicting age across held-out studies (median error: 2.65 weeks), and held-out cell types (median error: 2.86 weeks; Fig 3D). To further test the generalizability of this model, we applied it to two held-out hypothalamus datasets [22,27], obtaining accurate predictions in both (Correlation between actual and predicted ages, Herb: R = 0.9; Zhou: R = 0.89; Fig 3E). The cell type-agnostic model achieved the best performance across all cell types in held-out hypothalamus data (median error: 1.87 weeks), outperforming cell type-specific models (median error: 2.65 weeks, Wilcoxon test: $P=0.0069$, Fig 3F).

The robustness of this model comes from a diverse training dataset that broadly samples cross-cell type and cross-study variability. In contrast, models trained on data from only one study show good performance within that study (median error: 1.52 weeks) but fail to generalize to other studies (median error: 8.03 weeks, Wilcoxon test: $P=0.0078$). In sum, a meta-analytic cell type-agnostic model robustly predicts age across cell types and datasets of the developing human brain, eliminating the need for rigorous cell type alignment across datasets.

**Benchmarking model robustness.** We assessed model robustness to training data composition by varying the number of cells, studies, and anatomical diversity (S6 Fig). Even after controlling for number of cells in training (~1/10 of the full dataset), the cell type-agnostic model remained among the top two models for all cell types and maintained stable and comparable performance across cell types, whereas cell type-specific models performed well only within their own lineage. Increasing the number of training datasets improved performance, with gains plateauing after 6–7 datasets, indicating stable generalization well before all 13 available datasets were used. Prediction accuracy was lowest for third-trimester samples across all settings, reflecting limited sampling in this window rather than model failure. To assess anatomical generalization, we excluded cortical or hypothalamic datasets from training and evaluated performance in the held-out regions. Accuracy in cortical datasets was largely unaffected by exclusion of cortical data, whereas prediction in hypothalamic datasets was reduced when hypothalamic data were absent from training; this effect diminished as additional studies were included, even from other forebrain regions.

We further benchmarked the model against previously proposed developmental gene sets and nonlinear methods. Models based on maturation modules from He and Yu (2018) or birthdate-associated genes from Telley and colleagues (2019) [38,39] consistently underperformed relative to our approach, with higher error and greater inter-study variability (FDR-adjusted Wilcoxon $P<0.05$; S7 Fig). Across all approaches tested: regression models (cell type-agnostic, cell type-specific, compositional), nonlinear models (XGBoost, random forest, and MLP), and previous approaches, the cell type-agnostic model achieved the lowest error and highest correlation between predicted and actual age (S8 Fig).

Overall, we demonstrate that the cell type-agnostic model is robust to variation in training data, providing a practical and scalable approach for developmental staging across studies, brain regions, and experimental model systems. Next, we evaluated features used by the model, identifying gene sets that contribute to its robust performance.

## Sampling diverse neurodevelopmental programs drives robust predictions of cell type-agnostic model

The regularized cell type-agnostic model selects 462 genes (191 negative, 271 positive coefficients) that robustly predict developmental age across cell types and datasets (S3 Table). Feature importance analysis shows enrichment for global RNA regulatory processes, particularly m6A-RNA reader activity and mRNA stabilization (Fig 4A), involving *IGF2BP1* and *IGF2BP2* (S9 Fig). These conserved RNA binding proteins regulate mRNA stability, translation, and localization, and

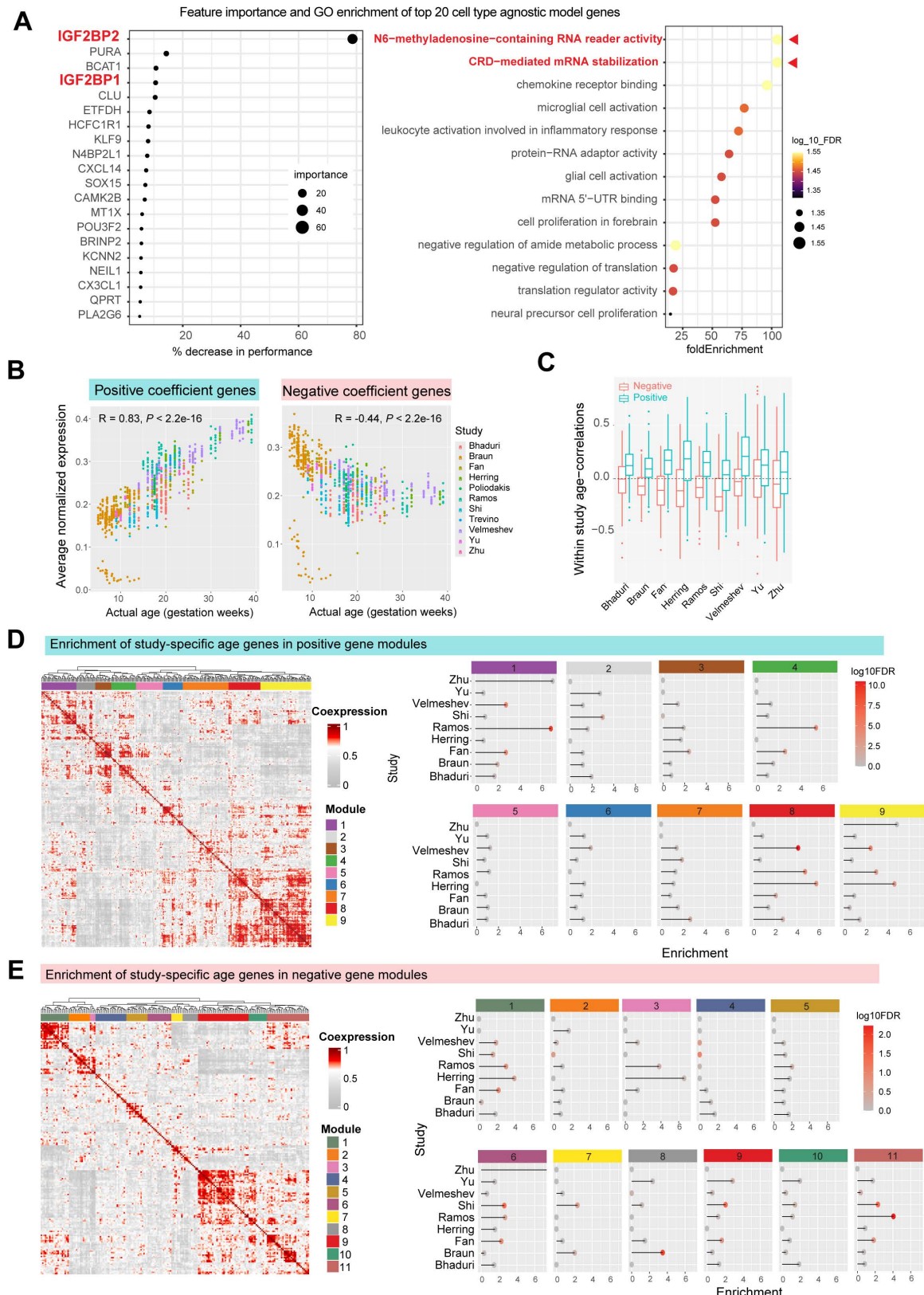

**Fig 4. Co-expression analysis of cell type-agnostic model genes reveals distinct contributions of study-specific developmental programs.**
A, Top 20 model genes ranked by their perturbation impact (left), GO enrichment highlights m6A mRNA reader activity as an enriched term, mediated

by *IGF2BP2* and *IGF2BP1* (right). **B**, Average expression of positive coefficient genes (left) and negative coefficient genes (right) across all datasets. **C**, Boxplot shows correlation coefficients of model genes with age within individual studies. Positive genes generally show higher correlation coefficients within every study compared to negative genes ($P < 10^{-3}$ for all studies, Wilcoxon test with FDR correction). **D**, **E**, Aggregate co-expression network of positive (D) and negative model genes (E) hierarchically clustered to identify distinct co-expression modules (9 positive and 11 negative modules). Panels on right show enrichment of study-specific age-correlated genes within each module. Each module represents age-genes from distinct study combinations.

play well-established roles in controlling the timing of neurogenesis and cortical layer formation across species [40–49] Additional high-ranking genes include regulators of neural stem cell proliferation (*BCAT1, PURA*) [50–52], mitochondrial metabolism (*ETFDH*) [53], amyloid processing (*CLU*) [54], transcription factors (*KLF9, SOX15, POU3F2*), and signaling genes (*CAMK2B*, *KCNN2*) involved in neural fate and maturation, indicating that the model draws on coherent, developmentally relevant programs. Importantly, filtering lowly expressed genes did not alter model performance or feature importance, indicating that these gene-level signals are robust to low-expression noise.

We compared top genes selected by cell type-specific models (S4 Table). To those used by the cell type-agnostic model (S10 Fig). Across all models, *IGF2BP1/2* consistently ranked among the top features, reflecting shared RNA regulatory programs of temporal progression across lineages, while cell type-specific models additionally leverage canonical lineage-restricted processes. These include cell cycle (progenitors), glial cell migration (astrocytes), regulation of reactive oxygen species and PI3K signaling (inhibitory neurons), glutamate transport (oligodendrocytes), and mesoderm morphogenesis (microglia). Notably, the cell type-agnostic model contained the fewest model-specific genes and showed the greatest overlap across cell types, consistent with its capture of global developmental programs rather than lineage-restricted signals.

At the expression level, aggregate expression of model genes with positive coefficients increased with gestational age (R = 0.83), while genes with negative coefficients decreased (R = −0.44, Fig 4B). Positive and negative model genes consistently retained their age-related expression trends within individual studies (S6 Table) and were clearly discriminable based on age-correlation (average AUROC = 0.74, Fig 4C). Notably, the model-selected genes are **not** the strongest age-correlated genes individually: a classifier trained on absolute correlation coefficients fails to distinguish them from the rest of the genome (AUROC = 0.54). We also examined the overlap between model genes and tissue-level PCs associated with age and found only weak enrichment (AUROC = 0.57 and 0.48 for positive and negative genes, respectively), indicating that our model captures cell-autonomous developmental signals that are largely distinct from tissue-level variation driven by cell type composition and technical effects.

Together, these results indicate that the cell type-agnostic model captures robust, shared temporal signals through the coordinated action of multiple biological processes, rather than reliance on single high-effect markers. To understand the biological basis of this predictive power, we conducted co-expression analysis to identify coordinated temporal gene modules in the model.

**Co-expression analysis identifies modular temporal gene programs.** We clustered positive and negative model genes into 20 co-expression modules and then evaluated the enrichment of study-specific age-genes in each module (Figs 4D and 4E and S5 Table). Our analysis revealed that each module consists of age-correlated genes from distinct studies, that represent diverse biological processes (Fig 5 and S7 Table). For example, among positive gene modules (Fig 4D), Module 4 is highly enriched for age-genes from Ramos and Fan datasets and represents amino acid transporter activity. Modules 8 and 9 are both highly enriched for genes from Velmeshev, Ramos, and Herring datasets, and represent glial cell proliferation and synaptic transmission, respectively. Modules 5 and 6 are not enriched for any one dataset but represent lipid catabolism and action potential-related genes, respectively (Fig 5A). Similarly, among negative gene modules (Fig 4E), Module 9 is enriched for Shi, Yu, and Fan datasets and represents neuropeptide signaling and GPCR

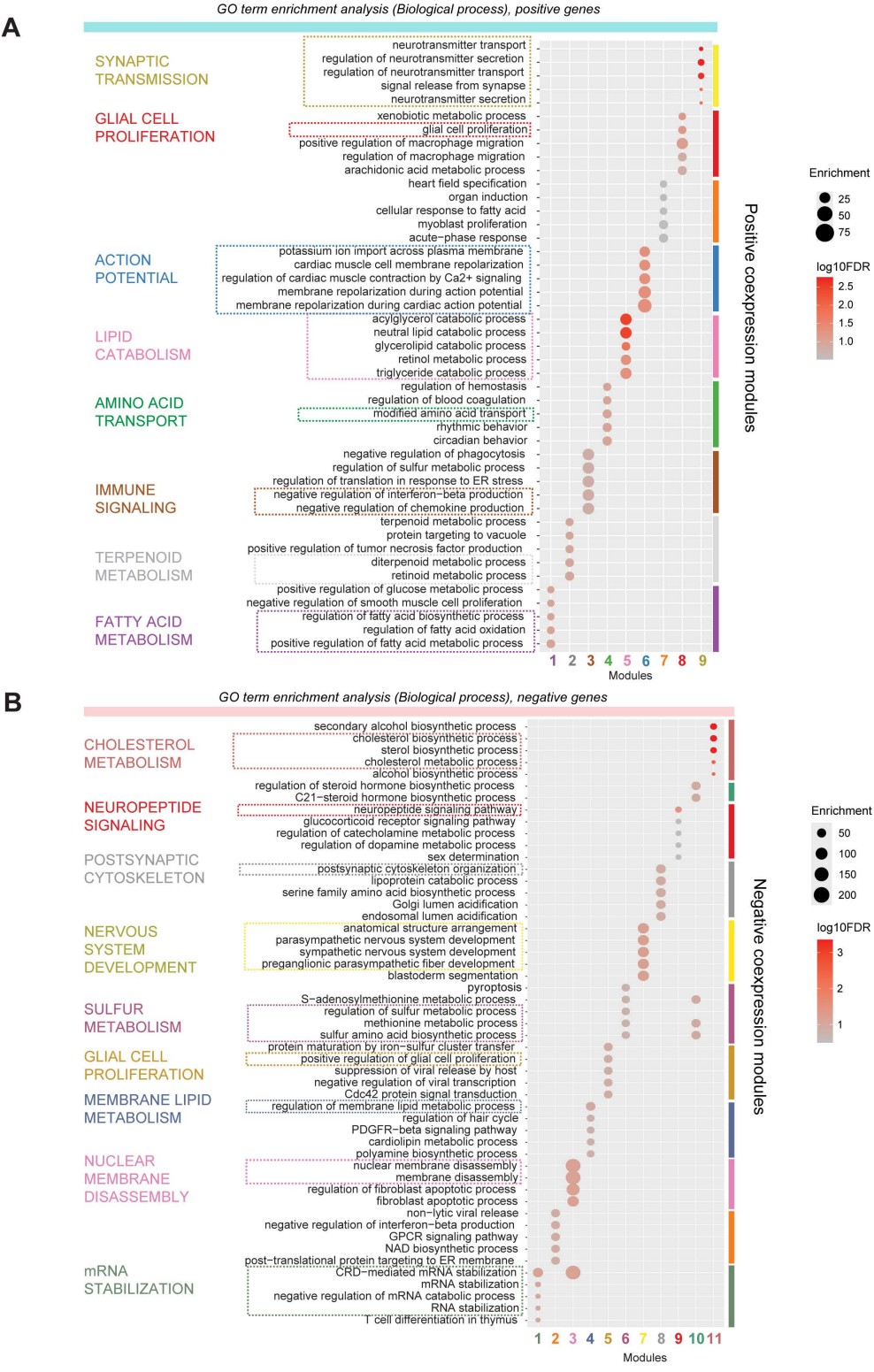

**Fig 5. GO enrichment of cell type-agnostic model genes highlights diverse biological processes contributing to developmental age prediction. A,** Positive gene modules are significantly enriched for distinct GO biological process terms including synaptic transmission, action potential, and lipid catabolism. **B,** Negative gene modules are significantly enriched for cholesterol metabolism, postsynaptic cytoskeleton, nervous system development, etc.

activity, while Module 11 is enriched for Shi, Ramos, and Fan datasets and represents cholesterol metabolism. Module 8 is specifically enriched for the Braun dataset and represents postsynaptic cytoskeletal organization (Fig 5B).

We next assessed the robustness and conservation of these modules. Within individual studies, all modules showed strong co-expression (mean AUROC = 0.71–0.91), and clustering studies by module co-expression clearly separated developmental stages, indicating that these modules track temporal milestones of fetal brain development (S11 Fig). To further examine whether modular co-expression reflects conserved biology, we used cross-species meta-analytic RNA-seq networks [55] to assess co-expression conservation. Compared to random gene sets, model genes showed significantly higher conservation of co-expression across a wide-range of mammals (S12A Fig, empirical *p*-values < 0.05 across 11 species spanning primates, rodents, rabbits, dogs, and ungulates). Together, these results indicate that the cell type-agnostic model selects genes that collectively form conserved, modular programs that encode developmental time across studies and are conserved across species.

As a complementary analysis, we systematically evaluated Gene Ontology (GO) terms by training regularized regression models restricted to genes from a single GO category (2,213 GO terms; Fig 6A). Models trained on synGO [56], a set of 1,112 synapse-related genes were also used for comparison. As expected, most GO-restricted models underperformed relative to models trained on the full gene set. Only a minority of GO terms outperformed the corresponding full gene, cell type-specific model, and this fraction varied widely by cell type (astrocytes: 31%, vascular cells: 19%, progenitors: 17%, oligodendrocytes: 13%, other: 8.1%, inhibitory neurons: 1%, excitatory neurons: 0.5%, and immune cells: 0.5%) Top-performing GO terms differed markedly across cell types (Fig 6B and S8 Table), resulting in poor cross-cell type concordance (Spearman correlation < 0.5; Fig 6C and 6D). Importantly, no GO-restricted model surpassed the performance of the cell type-agnostic model trained on all genes (empirical *P* = 0.00045; Fig 6A).

Overall, our work shows that developmental age prediction is driven by multiple biological processes. While cell type-specific accuracy can be enhanced by focusing on lineage-relevant pathways, robust cross-cell type prediction requires sampling diverse temporal programs: an advantage captured by the cell type-agnostic model.

Using our suite of developmental age predictors, we next probed whether these signals are preserved in other model systems of neurodevelopment. Specifically, we asked whether cell types in human neural organoids cultured in vitro recapitulate the tempo of cell-autonomous maturation seen in vivo.

## Cell type-agnostic model trained on primary tissue predicts developmental progression in human neural organoids

Human neural organoids are self-organizing 3D cellular aggregates derived from pluripotent stem cells, with the potential to recapitulate the cellular diversity of the developing human brain [57–60]. Organoids enable the characterization and manipulation of cell types that are otherwise inaccessible. While individual cell types in organoids are thought to align closely to the human brain [16,61,62], the extent to which lab-grown organoids reliably recapitulate the tempo of in vivo neurodevelopment is a topic of ongoing research [5,63,64]. High variability across batches, cell lines, and protocols has been reported previously [65]. We aimed to determine the developmental age of cell types in organoids with respect to the human brain to enable uniform comparisons.

Using our cell type-agnostic model, we assess developmental age in organoid cells across diverse protocols (Fig 7) without needing to integrate or align cell type annotations across datasets. Our model accurately predicts normal developmental progression in organoid cell types, supporting the preservation of in vivo molecular programs in cultured organoids. Predicted age across all cell types is highly correlated to in vitro culture age in both cortical (R = 0.91, Fig 7A) and midbrain organoids (R = 0.89) from Uzquiano and colleagues 2022, and Fiorenzano and colleagues 2022, respectively [16,66]. Furthermore, in a dataset where cultured organoids were transplanted into the rat brain [67], our model accurately predicts accelerated maturation in transplanted organoids (mean age = 32 weeks) compared to nontransplanted organoids (mean age = 24 weeks, $P < 2.2 \times 10^{-16}$). Within-sample variability of predicted developmental ages, computed as standard

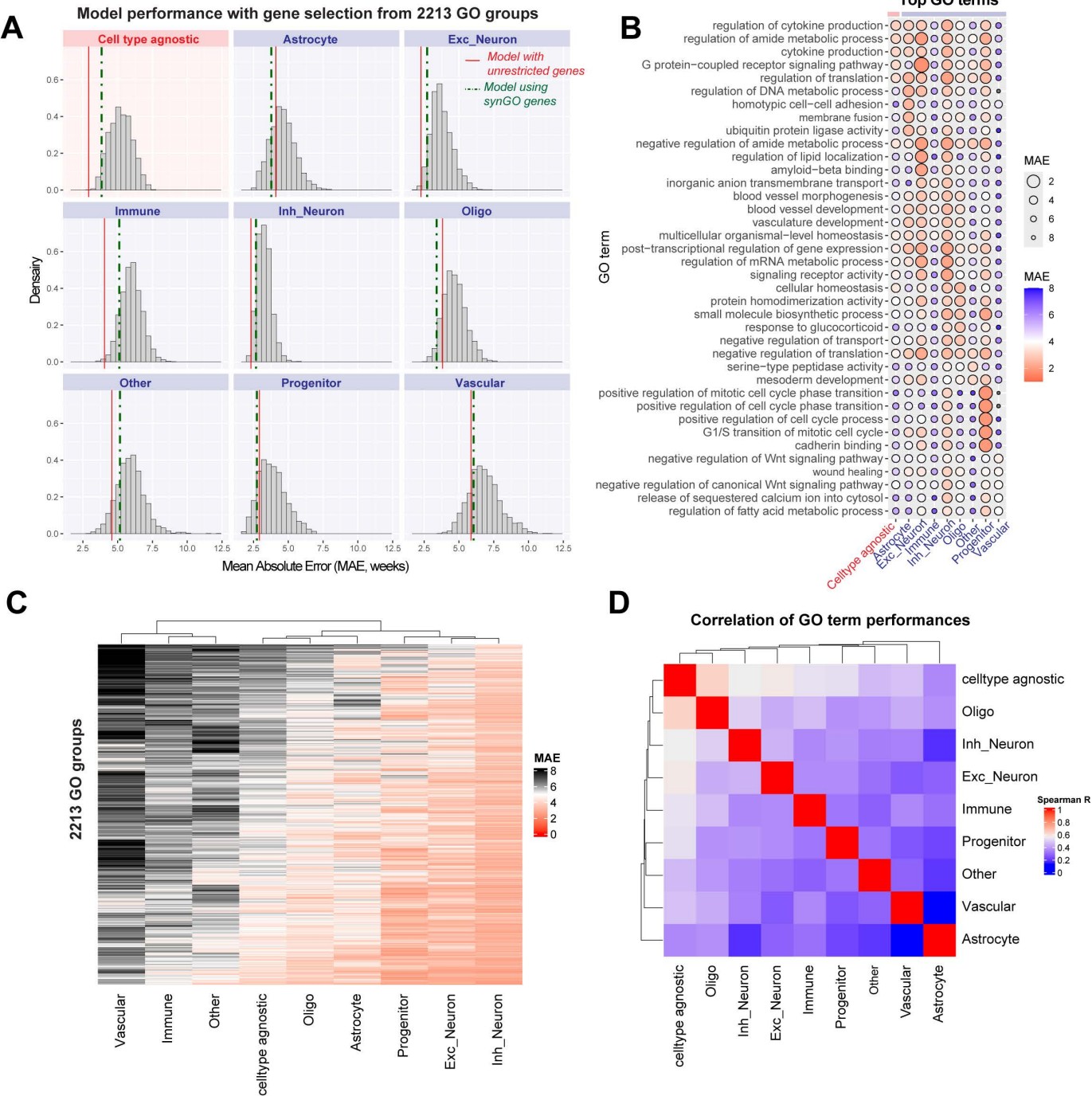

**Fig 6. Characterization of GO terms driving age prediction performance reveals cell type-specific signatures. A,** Age prediction models were trained using restricted gene sets derived from GO groups with 50–500 genes. Distribution of performances is shown for each of the cell type-specific and cell type-agnostic models. For comparison, red vertical lines indicate the performance of models trained using all genes (from Fig 3), and dashed vertical lines indicate performance of models using the synGO gene set. **B,** Dotplot shows the performance of top 5 best-performing GO terms in each cell type. In astrocytes: regulation of DNA metabolic process; progenitors: positive regulation of mitotic cell cycle phase transition; excitatory neurons: GPCR signaling pathway; inhibitory neurons: regulation of translation; oligodendrocytes: cellular homeostasis; immune cells: inorganic anion transmembrane transport; vascular cells: negative regulation of Wnt signaling pathway. **C,** Heatmap shows performance of all GO terms in each cell type. **D,** Spearman correlation of performance across all GO terms between cell types.

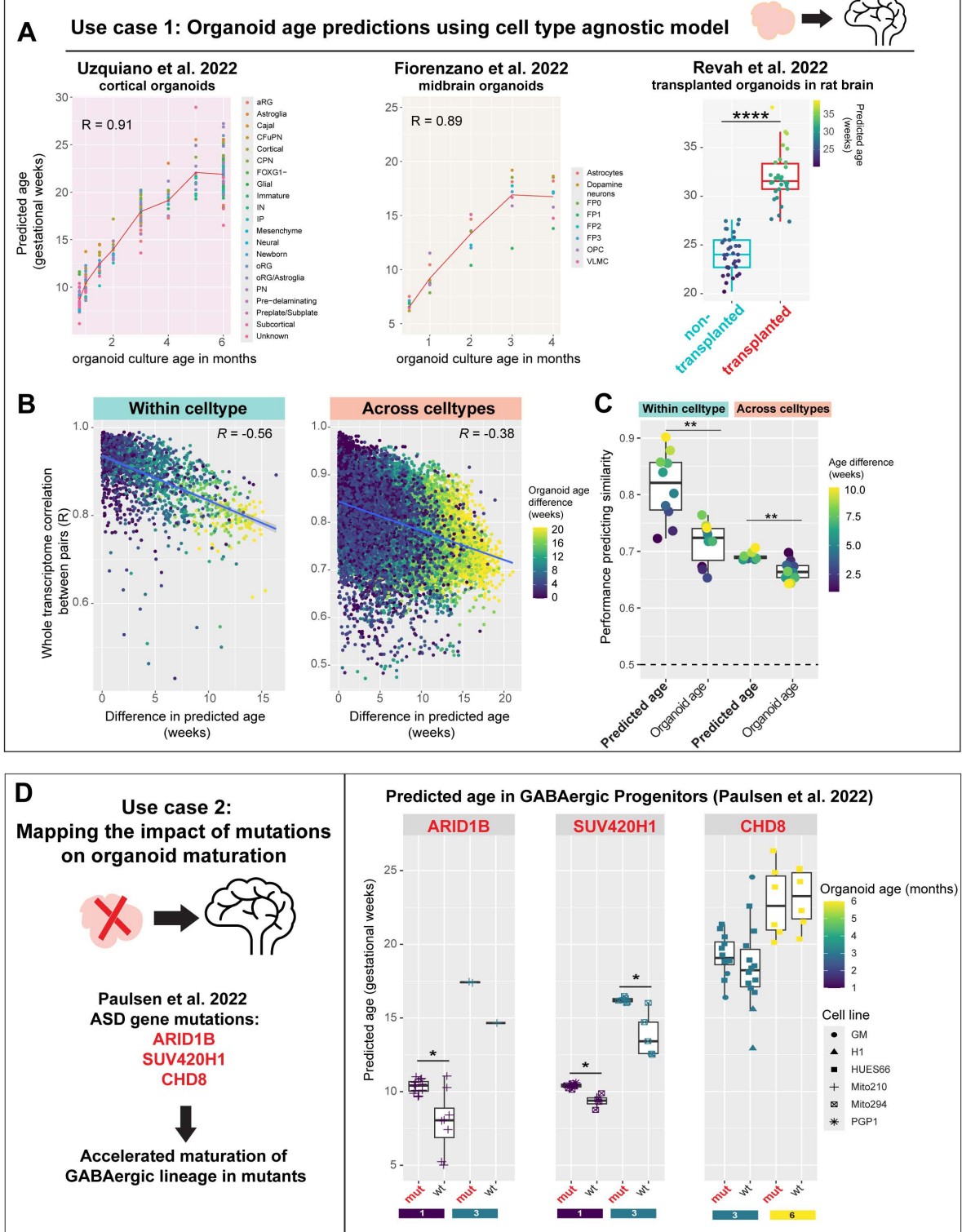

**Fig 7. Cell type-agnostic model predicts normal developmental trajectories and disease-associated shifts in human neural organoids. A**, Performance of the cell type-agnostic model (from Fig 3) in predicting developmental age in human neural organoid datasets from three different protocols. Predicted ages are highly correlated to the in vitro culture age of organoids in cortical (left, $P < 2.2e{-}16$) and midbrain organoids (middle, $P = 2.7e{-}14$). Cortical organoids transplanted into the rat brain show higher predicted ages compared to non-transplanted organoids (right, ****$P < 10^{-4}$, Wilcoxon test).

deviation of predictions per sample, is comparable between organoids and fetal samples (mean SD: 1.57 versus 2.08 weeks).

Independently, we assessed the relevance of predicted age in organoid cells by examining its ability to predict whole-transcriptome similarity. We expected that cells closer in developmental age would be more transcriptomically similar than cells further apart in age. Consistent with this, transcriptomic similarity between cell pairs is negatively correlated to the difference in their predicted ages, both within (R = −0.56, P < 2.2e−16) and across cell types (R = −0.38, P < 2.2e−16; Fig 7B). We also quantified this using an AUROC-based metric that tests whether cells closer in age are more transcriptomically similar across a range of age difference thresholds (Fig 7C). Across all thresholds (1–15 weeks), predicted age from the cell type-agnostic model consistently outperformed organoid culture age in predicting transcriptomic similarity; for example, at a 3-week threshold, AUROC = 0.77 versus 0.65. Overall, predicted ages show significantly higher performance in predicting transcriptomic similarity compared to organoid culture age both within (P = 0.0022) and across cell types (P = 0.0017, Wilcoxon test).

To explicitly assess variability in neural organoid maturation across protocols and batches, we compared co-expression strength of age prediction modules and fetal cell type MetaMarkers that measure organoid fidelity to primary tissue [64] (S12B Table). Age-module co-expression substantially varies across 24 organoid datasets (AUROC = 0.55–0.75) and is typically lower than cell type MetaMarker co-expression although the two measures are strongly correlated (R = 0.72, P < 2.2e−16). These results suggest that variability in maturation persists alongside broader variability in cell type marker fidelity across organoids.

**Detecting disease-related shifts and atlas-scale maturation in neural organoids.** We assessed whether our cell type-agnostic model can detect temporal shifts induced by disease-causing mutations, using cortical organoids with autism spectrum disorder (ASD)-linked mutations [68] (Fig 7D). Strikingly, the model predicts significantly older ages specifically in GABAergic progenitors of ARID1B mutants relative to wild type organoids at both 1 and 3 months of culture (mutant − wildtype age difference at 1 month: 2.40 weeks, P = 0.016; at 3 months: 2.76 weeks, n = 1). A similar pattern is also detected in SUV420H1 mutants (mutant − wildtype age difference at 1 month: 1.05 weeks, P = 0.0040; at 3 months: 2.37 weeks, P = 0.0086). In CHD8 mutants, we observe slightly older GABAergic progenitors in 3-month-old organoids from the HUES66 cell line (mutant − wildtype age difference: 0.88 weeks). These results align with the accelerated maturation of the GABAergic lineage in mutant organoids reported in the original study, which found a shift in the pseudotime trajectories of mutant GABAergic cells. Our model independently provides quantitative estimates for this age-shift in each cell type. Age predictions per cell line, culture age, and lineage for all organoids are shown in S13 Fig.

We further evaluated the robustness of our model in the Human Neural Organoid Cell Atlas (HNOCA), an integrated meta-atlas of 36 organoid single-cell datasets with 1.77 million cells from multiple protocols [69]. A challenge with large integrated datasets is the use of different technologies across studies, resulting in varying read depths that could affect age prediction from log-normalized expression. Therefore, we used a normalization-robust model variant that uses rank-normalized expression (see Materials and methods). This model successfully predicts developmental progression of cell types across all studies in the HNOCA atlas (S14 Fig), with overall correlation coefficient of 0.52 between the predicted age and organoid culture age. Consistent with known metabolic limitations of organoids grown in culture long-term [5], predicted ages decline after about 6 months in culture. Predicted age is most advanced by transplantation in vivo, as the oldest cell types in our organoid analysis are present in the rat-transplanted organoids from Revah and colleagues 2022.

**Inclusion of postnatal training data does not improve organoid maturation timeline.** To determine whether the apparent plateauing of predicted developmental age in neural organoids reflects biological constraints, limited training data, or model artifacts, we retrained a new cell type-agnostic model that extends the training set beyond fetal development by including postnatal human brain data up to adulthood from three datasets. This ensures that model predictions are not constrained by the upper bound of fetal ages in the original model. Applying this expanded model to in

**B**, Whole-transcriptome similarity of cell pairs (Pearson's R) is negatively correlated to their difference in predicted age both within and across cell types ($P < 2.2e−16$). **C**, Performance in predicting cell-cell similarity based on predicted age versus observed organoid age (days in vitro). $**P < 0.01$, Wilcoxon test. **D**, Developmental age prediction in cortical organoids with mutations in ASD-linked genes: ARID1B, SUV420H1, and CHD8. Right: predicted age of GABAergic progenitor cell types is significantly elevated in ARID1B and SUV420H1 organoids at both 1 and 3 months of culture ($*P < 0.05$, Wilcoxon test), matching predictions from the original study. CHD8 mutants show a minor increase in 3-month-old organoids.

vitro organoid datasets, we found no change in predicted ages compared to the fetal-only model (S15 Fig). Specifically, predicted ages continued to plateau at 5–6 months in cortical organoids (Wilcoxon test comparing original and revised predictions: FDR-adjusted $P = 0.47$) and at 3–4 months in midbrain organoids ($P = 0.80$).

In contrast, inclusion of postnatal training data significantly increased predicted ages only for the in vivo transplanted organoids. Median predicted age increases from 32 weeks with the fetal-only model to 77 weeks (~9 months postnatal) with the expanded model ($P = 2.32 \times 10^{−8}$). This selective effect demonstrates that the maturation plateau in organoids observed in vitro is not a computational artifact but likely reflects biological constraints which are alleviated by transplantation in vivo.

These results establish the generalizability of cell type-agnostic age predictors across diverse human neural organoid datasets. Our method enables uniform comparisons across organoid protocols, reveals lineage-specific temporal shifts caused by mutations, and extends to large integrated atlases. Next, we investigated whether models trained on human data could generalize to the embryonic mouse brain, to assess evolutionary conservation.

## Models trained on human brain tissue accurately predict accelerated pace of embryonic mouse brain development

The human brain develops at a slower pace compared to other species, a phenomenon called neoteny or bradychrony [70,71], whereas mice are thought to exhibit accelerated neurodevelopment. The mechanisms underlying differences in developmental tempo across species remain largely unknown. We aimed to compare cell-autonomous maturation across human and mouse neurodevelopment by using age predictors trained on human data, restricting to genes with mouse one-to-one orthologs.

First, we assessed whether cell-autonomous programs of human neurodevelopment were conserved in mice by evaluating the performance of a model trained purely on human data in an embryonic mouse brain dataset [72]. Strikingly, the cell type-agnostic model trained on human tissue accurately predicts developmental progression of cell types in the mouse brain—demonstrating strong conservation of cell-autonomous transcriptional timing across species (Fig 8A). Predicted human age is strongly correlated to embryonic mouse age in neurons ($R = 0.94$), neuroblasts ($R = 0.94$), glioblasts ($R = 0.9$), radial glia ($R = 0.9$), Vascular cells ($R = 0.94$), oligodendrocytes ($R = 0.76$), and immune cells ($R = 0.47$). However, very early nonneural cell types absent in the human data show poor performance, including endoderm ($R = −0.5$) and mesoderm cells ($R = 0.0084$). Intriguingly, we observe a transient decrease in predicted age around E8.5 in mouse neural progenitor cells, which might correspond to a similar "rejuvenation event" observed in the epigenetic DNA methylation age of mouse embryonic stem cells [73].

Next, we compared cell-autonomous maturation across the developmental span of human, mouse brains, and human neural organoids. By comparing the average predicted age of neurons, astrocytes and progenitors per sample against actual age (Fig 8B), we observe accelerated cell-autonomous maturation in mice compared to human brains and neural organoids (slope: mice = 9.762, humans = 0.811, human neural organoids = 0.550), with mice reaching maturation equivalent to 21 human weeks in just 18 days (ANOVA, effect of tissue: $F_{(2,249)} = 260.59$, $P < 2e−16$).

To assess the robustness of mouse-human tempo differences, we performed sensitivity analyses using alternative ortholog definitions and gene set restrictions. Models trained using one-to-one orthologs, orthogroups [74], random ortholog subsampling, or synapse-biased genes all yielded consistent cross-species performance and similar tempo estimates,

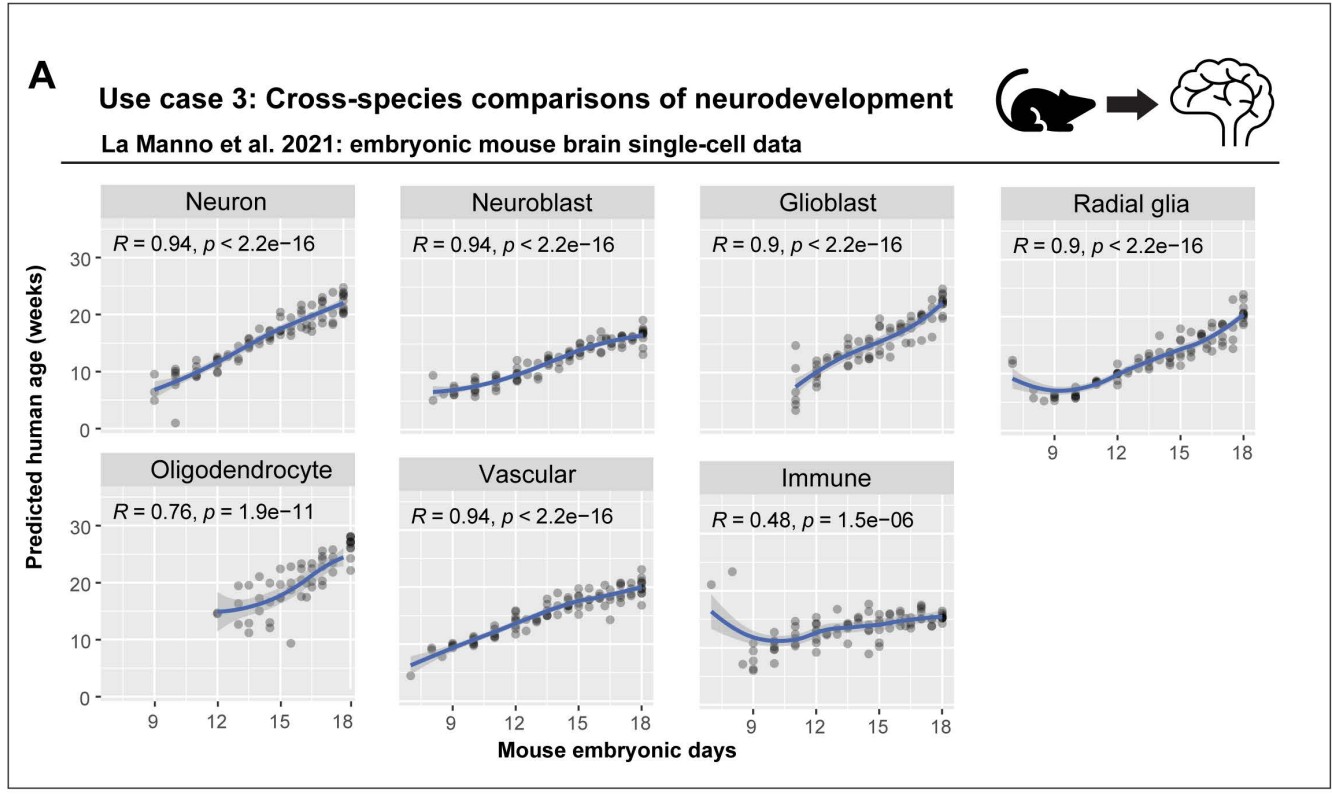

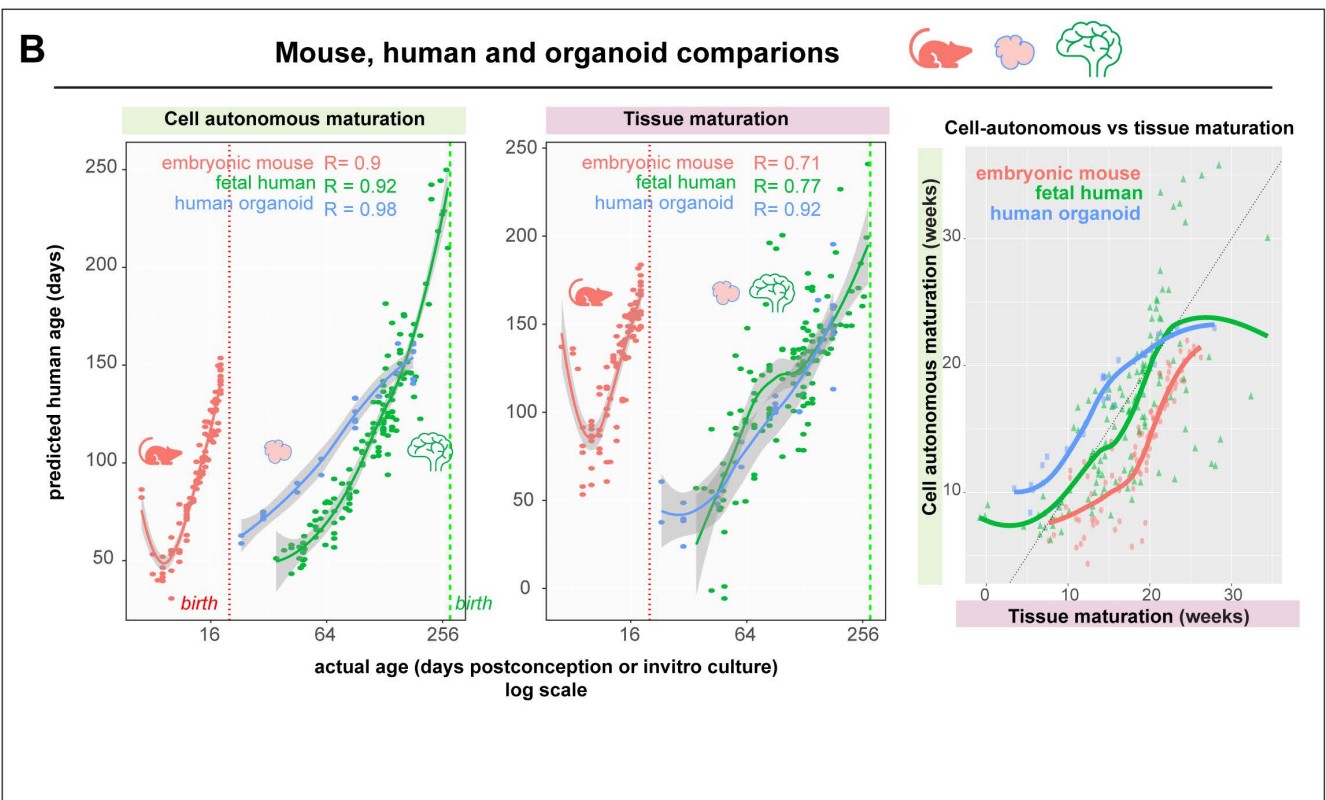

**Fig 8. Cross-species generalization of model trained on human brains enables uniform comparisons of neurodevelopment across mice, human brains and organoids. A**, Performance in predicting developmental age of cell types (in human weeks) in embryonic mouse brain development

using a cell type-agnostic model trained on human data. Genes were restricted to mouse orthologs. **B**, Comparing predicted cell-autonomous ages across mouse, human brain development, and human neural organoids (left). The middle panel shows tissue compositional age predicted from proportions of astrocytes and progenitors per sample in mouse, human brains, and human neural organoids. Average predicted cell-autonomous age of neurons, glia, and progenitors per sample is plotted against the compositional age, revealing a synchrony between cell-autonomous maturation and global tissue maturation across human, mouse brains, and organoids.

supporting ~8–12-fold acceleration in mouse development across gene set definitions. As an internal benchmark, we additionally trained a model exclusively on embryonic mouse brain data and evaluated it on human samples. Both within and across species, predicted ages are strongly correlated with chronological age (R = 0.93, 0.85). While the human model indicated accelerated mouse maturation, the mouse model conversely reveals slower human maturation (slopes ~0.02–0.07 across datasets). These results support robust cross-species ordering while indicating that absolute tempo estimates depend on dynamic range of the training data.

We further compared the tempo of tissue-level development using meta-analytic models based on astrocyte and progenitor proportions from Fig 2. This compositional model also predicts developmental progression in both organoids and mice brains, recapitulating cell-autonomous maturation tempo (slope: mice = 8.04, humans = 0.74, organoids = 0.58). Importantly, despite differences in the features used and their resolution, both tissue and transcriptome-based models show strong agreement (correlation in human: R = 0.69, $P < 2.2e−16$; mouse: R = 0.86, $P < 2.2e−16$; human neural organoids: R = 0.94, $P = 4.8e−13$). Overall, the tempo of cell-autonomous maturation and tissue-level development appear synchronized across systems (ANOVA, interaction between compositional age and model system, $F_{(2,249)} = 0.765$, $P = 0.466$). While these results require validation in additional datasets and species, they provide a framework to evaluate conserved, coordinated neurodevelopmental changes across model systems.

Finally, we show that the cell type-agnostic model also detects delayed maturation in hypoxic organoids [75] ($P = 0.035$), as well as tempo differences in interspecies midbrain organoids [76], with slowest tempo in humans, followed by chimpanzee and macaque cells (S16 Fig). Together, these results support the use of the cell type-agnostic transcriptomic model for downstream analyses requiring robust and sensitive developmental staging.

Our work establishes a foundation for comparing neurodevelopmental age across diverse contexts, model systems, and species. We provide a robust integration-free, cell type-agnostic age prediction model that will allow users to assess neurodevelopmental progression in their own single-cell datasets.

## Discussion

We analyzed 13 human fetal brain datasets with over 2.8 million cells to identify robust tissue-level and cell-autonomous predictors of neurodevelopmental age. We show that specific cell type proportions reliably predict tissue age across datasets and further train a cell type-agnostic model to track cell-autonomous maturation in diverse cell types. This model generalizes across multiple human brain, human neural organoid, and mouse brain datasets, revealing disease-specific and evolutionary shifts in neurodevelopmental tempo. Our work establishes a standardized metric for cross-dataset comparison of developmental time.

While individual studies have successfully measured developmental progression within their single-cell datasets [5,7,77], identifying robust patterns that generalize across multiple studies is challenging. Cross-study variability in cell type composition obscures tissue-level developmental changes, and known batch effects in single-cell data [78] further challenge the identification of generalizable cell-autonomous features of neurodevelopment. We show that the solution lies in meta-analysis: meta-analytic models trained on diverse datasets can extract highly reproducible tissue-level and cell-autonomous predictors of developmental age. Accounting for substantial cross-study variability enables robust developmental age prediction across human brain and neural organoid datasets, and most strikingly, demonstrates cross-species robustness.

A major advantage of our work is the prediction of developmental age across diverse cell types using a cell type-agnostic model. This is particularly relevant for stem cell-derived human organoid models where recent efforts have focused extensively on annotating in vitro cell types [69,79,80]. Comparing developmental age of cells across diverse organoid protocols has remained challenging due to the difficulty in matching cell types across protocols. By using a pre-trained cell type-agnostic model, we directly predict age in cell types across diverse neural organoid protocols, including the integrated HNOCA. This has the potential to identify interventions that alter maturation rates in vitro, as demonstrated by our prediction of lineage-specific shifts in organoids with ASD mutations [68]. Our framework could potentially also be leveraged beyond development to reveal selectively vulnerable and transitory cell populations with accelerated or delayed aging in aging atlases. Mechanistically, cell type-agnostic age prediction appears to involve conserved RNA regulatory programs—most prominently m6A-mediated mRNA regulation via *IGF2BP1/2* (also known as *Imp1/2)*, along with genes regulating stem cell proliferation, metabolism, and neuronal maturation among others.

A striking feature of the cell type-agnostic model is its ability to predict developmental progression in the embryonic mouse brain, despite being trained only on human brain tissue. Comparing predicted ages across human and mouse gestational span reveals greatly accelerated maturation during mouse brain development, in line with the known bradychrony of human neurodevelopment [81,82]. Exact quantitative estimates of developmental tempo vary depending on the process and timescale considered: smaller tempo differences between mouse and human (~2–5×) have been found in specific molecular and cellular programs in vitro (e.g., segmentation clock period in presomitic mesoderm [83], or protein stability in stem-cell derived spinal neurons [84]), whereas xenotransplantation studies have demonstrated much slower intrinsic maturation of human neurons in vivo [85]. Our estimate of a larger (~10-fold) difference emerges from a data-driven analysis of transcriptomic progression across the full human neurodevelopmental timeline, integrating diverse cell types and stages. Our results extend previous work by capturing global developmental timing differences across species.

In predicting developmental age from the cell's transcriptome, our models are also sensitive to the influence of the extracellular environment on developmental tempo. Notably, we accurately detect accelerated maturation in human neural organoids transplanted into the rat brain. In vitro models are a major avenue for studying cross-species differences in developmental tempo [83,85–87], and the extent to which the pace of development is set by cell intrinsic or extrinsic factors is a topic of ongoing research. Our framework to compare predicted tissue compositional age and cell-autonomous age across species and model systems may prove useful for quantitative assessment of interspecies differences [76]

We establish meta-analysis as an effective strategy for identifying conserved signatures in early neurodevelopment. This approach provides a standardized measure of developmental time, akin to previous aging metrics such as DNAm age [1,2,88,89] and transcriptomic aging clocks [36,37]. While incorporating additional datasets could further enhance robustness, our meta-analytic models provide a powerful framework for comparing developing brain cells across contexts, model systems, and species. These findings pave the way for optimizing in vitro models of neurodevelopment, uncovering disease mechanisms and advancing therapeutic interventions.

## Materials and methods

### Dataset download and pre-processing

Links for all downloaded datasets used in our meta-analysis are provided in S1 Table. A total of 13 human fetal brain datasets were used. Analysis was conducted using Seurat (v 5.1). Processed seurat objects for 7 human fetal brain datasets [24–27,29–31] and 2 human neural organoid datasets [16,67] were sourced from our previous meta-analysis [64]. For other datasets, processed data provided by the authors were used directly without additional filtering. Raw expression counts were log-normalized using Seurat's NormalizeData (normalization.method = "LogNormalize", scale.factor = 10,000) and used for all analyses unless otherwise specified. Genes were restricted to common genes found across all human fetal brain and neural organoid datasets resulting in a final set of 10,957 genes.

Author-provided metadata from the original publications were used to group cells by batch, age, and cell type. We mapped the author-assigned cell types to 7 broad cell types (Astrocyte, Progenitor, Neuron, Oligodendrocyte, Immune, Vascular, and Others; S2 Table). We validated our cell type groupings by estimating overlap of top 100 consensus marker genes per cell type across datasets (S1 Fig). Consensus markers for each author-assigned cell type were identified as recurrent marker genes across all samples in the study using Seurat's FindAllMarkers function, with criteria of $\log_2FC > 2$ and pct. $1 > 0.2$.

### MetaNeighbor assessment of cell type replicability

Python implementation of MetaNeighbor [35] (https://github.com/gillislab/pyMN) was used to quantify replicability of the 7 broad cell types across 9 datasets. Default settings were used, with fast_version = TRUE. AnnData objects were restricted to 212 highly variable genes identified by pyMN before running MetaNeighbor.

### Compositional age prediction models

**Study-specific compositional models:** Generalized linear models were trained to predict age (gestation weeks) of each sample within a study from the total proportions of each cell type present in that sample. Models were trained using Elasticnet regression (method = "glmnet") from the caret package (6.0.94) with leave one sample out cross-validation (trainControl method = "loocv"). Only studies with at least 4 different time points were used for this, resulting in a total of 11 study-specific compositional models (S3 Fig). Mean Absolute Error (MAE) was used to compare performance of each model within and across studies. Correlation coefficients and *p*-values shown are from the stat_cor function in ggpubr.

**PC analysis:** Raw expression counts within each sample were pseudo-bulked using Seurat's AggregateExpression (group.by = "orig.ident"). PCs were computed from pseudo-bulked expression matrices using Seurat's RunPCA (npcs = 5) within each dataset with more than 4 distinct time points. PC1 or PC2 were significantly correlated to sample age in each dataset (S4 Fig). A generalized linear model (GLM) was trained within each study to predict age using the PC most strongly correlated with age, resulting in a total of 8 study-specific models. Each study-specific model was tested on other datasets by projecting them onto the age-correlated PC used by the model and predicting age with the GLM (Fig 2C). MAE was used to compare performance of each model within and across studies.

**Meta-analytic compositional model:** We performed feature selection to identify cell types whose proportions reliably predicted age across studies. Meta-analytic linear models (method = "lm") were trained on 9 datasets to predict age in each held-out dataset (leave-study out cross-validation) using the proportions of individual cell types or combinations of 2 cell types. The best-performing cell type combination, astrocytes and progenitors, was identified to show least MAE across held-out datasets. Robustness of this astrocyte-progenitor meta-analytic model was tested further in two datasets (Herb and Zhou) that were held-out during training. *P*-values were estimated by comparing predictions of each model to the corresponding null model trained on data with randomly permuted ages.

### Cell-autonomous age prediction models

**Cell type-agnostic model:** Single-cell transcriptomes were aggregated by author-assigned cell type within each sample in a dataset using Seurat's AggregateExpression, resulting in a total of 905 transcriptomes from 11 datasets. A meta-analytic regularized regression model ("glmnet", caret package) was trained jointly on all cell types from all studies to use log-normalized gene expression to predict age. The gene set was restricted to genes present in all the datasets in our collection, resulting in 10,957 genes. This model uses elastic net regularization with mixing parameter ($\alpha = 0.1$) and the regularization strength ($\lambda = 0.3377$) selected by cross-validation to minimize mean absolute error on the training data. Performance was evaluated in held-out studies (leave-study out cross-validation) and held-out cell types (leave-cell type out cross-validation, Fig 3D). By nature of regularization, this model selects 462 genes out of 10,957 provided genes to

predict age robustly across cell types and datasets. Coefficients of genes used in the model are provided in S3 Table. The robustness of this cell type-agnostic model was further validated in two datasets (Herb and Zhou) that were held-out during training. This model is the one used to evaluate developmental progression in external human neural organoid datasets (Fig 7).

**Cell type-specific models:** We trained cell type-specific models on the transcriptomes of each broad cell type (Excitatory neurons, Inhibitory neurons, Oligodendrocytes, Progenitors, Astrocytes, Vascular, Immune, and Others). All cell type-specific models were trained using the same elastic net regression framework (glmnet) using log-normalized gene expression of the full gene set (10,957 genes), with hyperparameters selected by cross-validation within each model. Coefficients of genes used by these models are provided in S4 Table. These models showed similar performance on their training cell type (MAE in cross-validation) and held-out cell types, indicating limited cell type-specificity (Fig 3C). Since their performance was lower than the cell type-agnostic model, they were not used for predictions on external datasets.

**Benchmarking model robustness:** To assess robustness to training data composition, we performed systematic subsampling and retraining of cell-autonomous age prediction models while varying (i) the number of cells, (ii) the number of contributing studies, and (iii) anatomical coverage of the training set (S6 Fig). For cell-number control, 100 cell type-specific and cell type-agnostic models were each retrained on random subsets ($n = 30$, 50, and 70 cells) of the full training dataset. To evaluate dependence on study diversity, models were trained on all combinations of 2–13 datasets and tested on held-out studies. Developmental stage–specific performance was assessed by stratifying test data into trimester bins. Anatomical generalization was examined by excluding cortical or hypothalamic datasets from training and evaluating prediction accuracy in the corresponding held-out regions.

For method benchmarking, we retrained LASSO models (train method = "lasso") using previously published developmental gene sets [38,39] (S7 Fig). We additionally trained non-linear models (XGBoost, method = "xgbTree"; random forest, method = "rf"; and multi-layer perceptron with 10 hidden units, method = 'mlp') using identical input features and cross-validation splits. Model performance was quantified using MAE and correlation with chronological age. Summary of performance comparison across all approaches reports median performance values across cross-validation folds (S8 Fig).

**Feature importance analysis:** Feature importance was assessed using two complementary metrics: permutation importance and feature stability [90,91]. For permutation importance, the expression values of each gene were independently permuted across samples, and the resulting decrease in age prediction performance was quantified. Feature stability was estimated as the frequency with which each gene was selected across models trained on repeated subsamples of the training data (S10 Fig).

## Aggregate co-expression network and module analysis

To characterize gene sets contributing to the cell type-agnostic model's performance, we constructed aggregate co-expression networks for the positive and negative coefficient genes used in the model. First, we constructed a co-expression matrix per dataset using Spearman correlation coefficient between genes calculated from the cell type-level aggregated transcriptomes. The correlation coefficients were then ranked and divided by the maximum rank to obtain a rank-standardized co-expression matrix. Average rank-standardized co-expression across datasets ($n = 11$ used in model training) was computed for each gene-pair to generate the aggregate co-expression network for a set of genes. The aggregate co-expression network was hierarchically clustered using hclust and cut at a specified height using cutree ($h = 4.5$, or $h = 3$ for positive and negative gene modules, respectively) to generate 9 positive and 11 negative gene modules. The genes corresponding to each module are listed in S5 Table.

**Co-expression strength and conservation analysis:** We used EGAD [92] to quantify co-expression strength of age prediction gene modules within networks constructed from individual studies (https://github.com/sarbal/EGAD). EGAD applies 3-fold cross-validation, iteratively withholding genes from a module and testing whether their membership can be recovered based on network connectivity, with performance summarized as an AUROC score. To assess cross-species

conservation of co-expression, we used CoCoCoNet [55] to access meta-analytic RNA-seq co-expression networks across mammals (https://cocoblast.ccbr.utoronto.ca/) and tested whether the co-expression structure of model genes was more conserved between human and other species than expected by chance.

**Study-specific age-correlated genes:** We identified age-correlated genes within each study by calculating the Pearson correlation coefficient and *p*-value (cor.test) between each gene's log-normalized expression from cell type-aggregated transcriptomes and sample age. *P*-values were corrected for multiple hypothesis testing using the false discovery rate (FDR) method with p.adjust (method = "fdr"). Statistically significant age-correlated genes (adjusted $P < 0.05$) from each study are listed in S6 Table. Enrichment of age-correlated genes from each study in the cell type-agnostic model (Fig 4) was calculated using Fisher's exact test with FDR correction.

## GO term enrichment and characterization

**Enrichment of GO terms in co-expression modules:** GO annotations were obtained from GO.db and org.Hs.eg. db (sourced 2024-01-17). The hypergeometric test (phyper) was used to test the enrichment of specific GO terms in each co-expression module. *P*-values were corrected for multiple hypothesis testing using the FDR method with p.adjust (method = "fdr"). Only GO terms with 10–200 genes were evaluated. Results for GO Biological Process terms are shown in Fig 5. Enrichment of GO Biological Process, Cellular Component, and Molecular Function terms in positive and negative co-expression modules is listed in S7 Table.

**Age prediction performance of specific GO terms:** To further explore gene sets driving age prediction performance across datasets, we trained cell type-agnostic and cell type-specific models using regularized regression (caret package, method = glmnet) with genes restricted to specific GO terms. In total, 2,213 GO terms with 50–500 genes were tested. As before, cell type-level aggregate transcriptomes were used for training the models. The performances (MAE) of 2,213 GO terms for each of the cell type-specific and cell type-agnostic models are shown in Fig 6 and listed in S8 Table. For comparison, models were also trained using the synGO gene set with 1112 genes [56]. Spearman correlation coefficient of performance was computed between pairs of cell types across all 2,213 GO terms to assess cell type-specificity of GO terms in age prediction (Fig 6D).

## Developmental age predictions in external datasets

**Organoid age predictions:** Single-cell transcriptomes from human neural organoid datasets (Uzquiano and colleagues, Fiorenzano and colleagues, Revah and colleagues) were aggregated by author-assigned cell types within each sample using Seurat's AggregateExpression. The pre-trained cell type-agnostic model (Fig 3) predicted ages for each dataset using the predict function from the caret package. Predicted human gestational ages of cell types per organoid sample were compared to organoid in vitro culture age (Fig 7A). In the Paulsen and colleagues dataset, age predictions by the cell type-agnostic model were used to assess differences in the GABAergic lineage between wildtype and ASD-mutant organoids (*ARID1B, SUV420H1,* and *CHD8*). Age differences for all lineages by cell line, gene, and organoid age are shown in S13 Fig.

**Assessing whole-transcriptome similarity with predicted age:** Spearman correlation coefficients between cell type-aggregated transcriptomes were computed within each organoid dataset (Uzquiano and colleagues, Fiorenzano and colleagues, Revah and colleagues), producing a cell-cell similarity matrix based on all genes. To evaluate performance in predicting cell-pair similarity based on age differences, we ranked the correlation coefficients and compared the average rank of pairs with age difference below and above a certain threshold. This results in the Mann–Whitney *U*-statistic or AUROC, which we computed separately for predicted age and actual organoid age, in cell pairs of the same cell type and different cell types.

**Age prediction in integrated human neural organoid cell atlas:** To support generalization of the cell type-agnostic model to large integrated atlases with varying read depths, we trained a normalization-robust model that uses rank-normalized data instead of log-normalized expression. Here, each gene's raw count is replaced by its rank relative to other genes within that cell and divided by the maximum rank to obtain rank-normalized expression values within each dataset.

The integrated HNOCA h5ad object [69] comprising 36 organoid single-cell datasets with 1.77 million cells was downloaded from CellxGene. We aggregated expression at the cell type-level within each dataset and batch, using grouping variables based on 'publication', 'batch', 'cell_type_original', 'organoid_age_days', 'state_exact', and 'annot_level_1'. This resulted in a total of 9,811 transcriptomes, which were rank-normalized and then used for age prediction with the normalization-robust cell type-agnostic model. Predicted ages across all 36 datasets were compared to organoid in vitro culture age (S14 Fig).

**Quantifying organoid variability with co-expression analysis:** To quantify variability in developmental age signals across organoid preparations, we assessed co-expression of age prediction gene modules across 24 organoid scRNA-seq datasets (173 organoid batches) [64]. Module co-expression strength was computed using EGAD, and compared against co-expression of established fetal cell type MetaMarker gene sets that predict organoid fidelity to primary tissue (https://github.com/JonathanMWerner/preservedCoexp).

**Age predictions in the embryonic mouse brain:** The embryonic mouse brain dataset from La Manno and colleagues with 292,465 cells from 93 samples was downloaded as a loom file and converted to a Seurat object. We aggregated expression at the cell type-level within each sample, resulting in a total of 1,177 transcriptomes. We predicted age in the mouse brain using a variant of the cell type-agnostic model trained purely on human brain tissue, but with genes restricted to mouse orthologs present in the La Manno dataset ($n = 9,460$ genes). One-to-one orthologs between human and mouse were downloaded from http://www.ensembl.org/biomart/martview. Predicted human age was obtained using log-normalized expression of one-to-one mouse orthologs in the embryonic mouse brain dataset and compared to the actual mouse age (embryonic days).

**Alternative ortholog sets for human to mouse predictions:** To assess robustness of cross-species developmental tempo estimates to gene set choice, we used three complementary approaches. (i) Leveraging gene family definitions from OrthoDB v12.1 (accessed Mar 21, 2025), age prediction models were trained using all orthogroups ($n = 8,117$) shared between human and mouse (with expression averaged across genes per orthogroup), or specifically orthogroups with multiple genes (many to many orthogroups, $n = 1,185$). (ii) To test sensitivity to lineage-biased gene modules, we retrained 20 models using random subsets of 1,000 one-to-one orthologs and quantified variability in inferred tempo slopes. (iii) We explicitly tested synapse-biased gene sets by training models either restricted to SynGO genes or excluding SynGO genes from the ortholog set. All models were evaluated using leave-study-out cross-validation in human data and tested for cross-species generalization in the La Manno and colleagues mouse dataset, with developmental tempo inferred from linear fits between predicted and actual age.

**Mouse embryonic age prediction model:** For reciprocal mouse to human comparisons, we trained a meta-analytic cell type-agnostic age prediction model exclusively on embryonic mouse brain single-cell datasets ($N = 10$ studies, S9 Table), restricting genes to one-to-one orthologs shared across human and mouse datasets (9,615 genes). We evaluated cross-species generalization by applying the mouse-trained model to human data, quantifying correlations and linear slopes between predicted and chronological age.

## Compositional versus cell-autonomous predictions

Using predictions from the cell type-agnostic model, we compared the average age of astrocytes, progenitors, and neurons per sample to the actual age of the tissue (embryonic/in vitro days) across mouse brain, Uzquiano and colleagues organoid, and fetal human brain datasets. Similarly, compositional age for each organoid, human, or mouse brain sample was predicted using the astrocyte-progenitor meta-analytic model and compared to the actual age of the sample. Slopes were computed for each of the curves by fitting a simple linear model. Predicted compositional and cell-autonomous ages for each sample were also compared to each other and ANOVA was used to estimate the influence of tissue on developmental dynamics.

## R and R packages

All analysis were carried out in R version 4.4.1. Colors for heatmaps were selected using RColorBrewer and circlize. All plots were made using ggplot2 (v 3.5.1). Heatmaps are plotted using ComplexHeatmap [93]. Cell type-agnostic age

prediction models are made available at https://github.com/sridevi96/NeuroDevTime (RRID: SCR_026651). Data and code used to generate figures are available at Zenodo (https://doi.org/10.5281/zenodo.14908185).

## Supporting information

**S1 Fig. Overlap of consensus marker genes in cell types across human brain datasets.** Author-provided cell type annotations from each study were grouped into 7 broad cell types (Astrocyte, Progenitor, Neuron, Oligodendrocyte, Immune, Vascular, and Others; see S2 Table). Heatmap shows the overlap of top 100 consensus marker genes for each cell type across datasets. Cell classes show consistent overlaps in consensus markers across datasets, except for "Other" cells which include multiple different cell types.
(TIF)

**S2 Fig. MetaNeighbor assessment of cell type replicability in fetal brain single-cell datasets. A,** Schematic illustrating MetaNeighbor procedure: cell similarity networks constructed from gene-gene correlation are used to assess cell type replicability with a leave-study out cross-validation approach. The AUROC score measures whether cells of the same type rank higher than other cell 4 types in the network. **B,** Heatmap shows MetaNeighbor AUROC scores for all versus all comparisons of major cell classes across 9 datasets used for model training. Cells cluster by cell type rather than study **C,** Heatmap shows best versus next best MetaNeighbor AUROC scores measuring the ability to discriminate cells of the same type from the next closest cell type. Cells cluster by cell type rather than study.
(TIF)

**S3 Fig. Study-specific compositional model performance.** Performance of regularized regression models trained to predict sample age from cell class proportions within each study. Study-specific compositional models accurately predict gestational age in 7 out of 11 studies, with significant correlation between predicted and actual ages.
(TIF)

**S4 Fig. Principal component age predictions are primarily driven by cell type composition. A,** PC1 or PC2 scores are strongly correlated to age within each of the 8 datasets. Size of dots represents the proportion of progenitor cells in the sample. Percentage of variance explained by the PC is shown on the y-axis label. **B,** Correlation of PC with age is plotted against correlation of the PC with proportions for each cell type. Age-PC correlations are perfectly matched by PC-cell type proportion correlations for most cell types across all datasets. Shape of points indicates which principal component, and color represents the PC that was used for age prediction in panel A.
(TIF)

**S5 Fig. Sensitivity of Principal component age predictions to cell type removal.** Panels show the absolute correlation coefficient between principal components (PC1–PC5) and age in each dataset. PCs were computed from pseudo-bulked gene expression. Colors indicate PCs recomputed after removing all cells of a specific cell type from the dataset. Removing cell types shifts the PC-age-correlation in 4/8 datasets shown in the top row. Red arrows indicate complete change in the PC-age-correlation: i.e., in Braun and colleagues, removing Progenitors from the dataset shifts the age-correlation from PC2 to PC1. In Zhou and colleagues, removing Progenitors or Neurons from the dataset shifts age-correlation from PC2 to PC1.
(TIF)

**S6 Fig. Benchmarking performance of cell type-agnostic model to training data size and composition. A,** Boxplots show distribution of prediction error (Mean Absolute Error, MAE) in test data for 100 cell type-specific and cell type-agnostic models that were trained on the same number of cells ($n = 70$). **B,** Distribution of prediction error for models trained on different combinations of studies ($N = 2$–12). Performance increases as more datasets are used for training,

but plateaus after ~7 studies. Third-trimester samples show highest prediction error due to scarcity of training data. **C,** Distribution of prediction error in test datasets from cortex (top) or hypothalamus (bottom) when these brain regions are included or excluded from training data. ***$P<0.001$, **$P<0.01$ (FDR-adjusted Wilcoxon *p*-value)
(TIF)

**S7 Fig. Benchmarking performance of cell type-agnostic model against maturation modules from He and Yu 2018 and Telley and colleagues 2019.** L1 (LASSO) regularized regression models to predict age were trained using neuronal maturation gene sets from He and colleagues 2018 and Telley and colleagues 2019. Boxplots compare prediction accuracy (mean absolute error, MAE) in each of the human fetal datasets to the cell type-agnostic model developed in this study. (* adjusted Wilcoxon $P<0.05$).
(TIF)

**S8 Fig. Summary comparison of all models highlights cell type-agnostic model as top performer.** Plots show median correlation and mean absolute error (MAE) in cross-validation for each of the models on the x-axis. Venkatesan-celltypeagnostic, Venkatesan-celltypespecific, and Venkatesan-compositional are models developed in this paper. MLP, xgboost, and randomforest are more complex nonlinear models trained on our data. Lasso-He2018 and Telley2018_birthdate_lasso are models trained on our data but using gene sets derived from previous studies as external comparisons.
(TIF)

**S9 Fig. Expression of top cell type-agnostic model genes: *IGF2BP1, IGF2BP2, BCAT1*, and *PURA*.** Plots show average log-normalized expression of 4 cell type-agnostic model genes over age in different cell types. Each gene shows distinct and dynamic expression trends across cell types. *IGF2BP1/2* are generally negatively correlated to age, *PURA* is positively correlated, and *BCAT1* shows variable expression trends across cell types and developmental windows.
(TIF)

**S10 Fig. Top genes used by cell type-specific and cell type-agnostic models and their overlap. A,** Top 20 genes per model identified by feature stability: i.e., recurrent usage across 100 models trained on resampled data. **B,** UpSet plot showing overlap of top 100 model genes across cell type-specific and cell type-agnostic models. The red arrow points to smaller number of unique genes used by the cell type-agnostic model compared to cell type-specific models.
(TIF)

**S11 Fig. Co-expression modules from model genes show dynamic expression and track development. A,** Dynamic expression trends of each module from the 462 cell type-agnostic model genes. Plots show normalized average expression of genes per module. **B,** Schematic of EGAD pipeline to determine co-expression strength of a gene set. **C,** Correlation of module EGAD scores between studies groups datasets by age. Studies sampling similar temporal windows group together.
(TIF)

**S12 Fig. Evolutionary conservation and variability in organoid co-expression of developmental programs. A,** Cross-species co-expression conservation of cell type-agnostic model genes measured by −log10 (FDR) compared to random gene sets. Model gene co-expression is highly conserved across mammals. **B,** Correlation of developmental module co-expression scores from our cell type-agnostic model and fetal cell type MetaMarker co-expression from Werner and Gillis 2,024 in multiple organoid datasets. Developmental modules show generally lower co-expression than cell type modules, but both are tightly correlated across 24 organoid datasets.
(TIF)

**S13 Fig. Cell-autonomous age predictions in ARIDB1, CHD8, and SUV420H1 mutant organoids from Paulsen and colleagues 2022.** Predicted ages in each cell type from Paulsen and colleagues 2022 neural organoids separated

by gene, cell line, and organoid culture age (in months). Dotted boxes highlight GABAergic cell types where mutant cells show accelerated maturation, as per the original study.
(TIF)

**S14 Fig. Developmental age predictions in integrated human neural organoid cell atlas (HNOCA). A,** UMAP of HNOCA cells taken from CellxGene browser (left). The right panel depicts cross-validation performance of cell type-agnostic model trained on rank-normalized expression values from human fetal brain cell types. **B,** Developmental age predictions for organoid cells from different studies in the integrated HNOCA dataset. Predicted age is overall strongly correlated to organoid culture age in vitro. Square points show mean predicted age of all cell types per time point in each study.
(TIF)

**S15 Fig. Including postnatal data in model training extends age predictions only for transplanted organoids. A,** Cross-validation performance of cell type-agnostic age prediction model trained on both prenatal and postnatal data from 11 datasets. Predicted ages are strongly correlated with actual age (weeks since conception). Boxplots show predicted ages grouped by developmental windows: prenatal (<40 gestation weeks), infancy (<1year postnatal), childhood (<10 years), adolescence (<20 years), and adulthood (>20 years). **B,** Developmental age predictions for neural organoids using the original cell type-agnostic model trained on prenatal data only (red) and the updated model trained on prenatal plus postnatal data (blue). In vitro organoids show a plateau in predicted age at ~20 gestational weeks with both models, whereas transplanted organoids exhibit substantially extended maturation with the updated model, reaching predicted ages of up to 1–2 years postnatally (****$P < 10-4$, ns: not significant, FDR-adjusted Wilcoxon $P$-value)
(TIF)

**S16 Fig, Cell type-agnostic model detects hypoxia-linked maturation defects and species-specific rates of maturation in organoids.** (Top) Cell type-agnostic age predictions show significant delay in maturation in neural organoids grown in hypoxia compared to hyperoxia in dataset from Umans and Gilad 2025. (Bottom) Species-specific rates of maturation measured by the slope between predicted and actual age in interspecies midbrain organoids from Nolbrant and colleagues 2024. Slope and 95% CI are shown as error bars. Human cells show slowest rate of maturation while other primates are faster. *$P < 0.05$, **$P < 0.01$, $t$ test comparing slopes from linear model.
(TIF)

**S1 Table. Excel file with information on human datasets used for meta-analysis.**
(XLSX)

**S2 Table. Excel file with cell type mapping.**
(CSV)

**S3 Table. Excel file with coefficients of cell type-agnostic model, related to** Fig 3.
(CSV)

**S4 Table. Excel file with coefficients of cell type-specific models, related to** Fig 3.
(XLSX)

**S5 Table. Excel file with list of study-specific age-correlated genes, related to** Fig 4.
(XLSX)

**S6 Table. Excel file with co-expression modules derived from model genes related to** Fig 4.
(CSV)

**S7 Table. Excel file with GO enrichment results of co-expression modules related to** Fig 5.
(XLSX)

**S8 Table. Excel file with performances of GO term models related to Fig 6.**
(CSV)

**S9 Table. Excel file with information on mouse datasets used for model training.**
(XLSX)

## Author contributions

**Conceptualization:** Sridevi Venkatesan, Jesse Gillis.

**Data curation:** Sridevi Venkatesan, Jonathan M. Werner.

**Formal analysis:** Sridevi Venkatesan.

**Funding acquisition:** Yun Li, Jesse Gillis.

**Investigation:** Sridevi Venkatesan, Jesse Gillis.

**Methodology:** Sridevi Venkatesan.

**Project administration:** Jesse Gillis.

**Resources:** Jesse Gillis.

**Supervision:** Yun Li, Jesse Gillis.

**Writing – original draft:** Sridevi Venkatesan.

**Writing – review & editing:** Sridevi Venkatesan, Jonathan M. Werner, Yun Li, Jesse Gillis.

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
