## [Editor Report · Decision Letter 0]

22 Jul 2025

Dear Dr Gillis,

Thank you for submitting your manuscript entitled "Cell Type-Agnostic Transcriptomic Signatures Enable Uniform Comparisons of Neurodevelopment" for consideration as a Research Article by PLOS Biology.

Your manuscript has now been evaluated by the PLOS Biology editorial staff, and I am writing to let you know that we would like to send your submission out for external peer review.

Once your full submission is complete, your paper will undergo a series of checks in preparation for peer review. After your manuscript has passed the checks it will be sent out for review. To provide the metadata for your submission, please Login to Editorial Manager (https://www.editorialmanager.com/pbiology) within two working days, i.e. by Jul 24 2025 11:59PM.

Kind regards,

Taylor

Taylor Hart, PhD,

Associate Editor

PLOS Biology

thart@plos.org

---

## [Decision Letter · Decision Letter 1]

23 Sep 2025

Dear Dr Gillis,

Thank you for your patience while your manuscript "Cell Type-Agnostic Transcriptomic Signatures Enable Uniform Comparisons of Neurodevelopment" was peer-reviewed at PLOS Biology. It has now been evaluated by the PLOS Biology editors, an Academic Editor with relevant expertise, and by several independent reviewers.

In light of the reviews, which you will find at the end of this email, we would like to invite you to revise the work to thoroughly address the reviewers' reports.

As you will see, the reviewers indicate that your study addresses an important problem. However, they raised concerns about missing information and posed important questions about the methodology, analysis, biological significance, and use cases of the approach. We think that the advance from your study is primarily methodological, and we would like to consider it for our Methods & Resources category going forward; please revise your article accordingly and select this article type when you submit your revision. In your revision, you should thoroughly respond to all of the reviewers' comments, particularly Reviewer 1's point 4. In addition, we strongly encourage you to benchmark your study against other available methods.

Here is a brief description of what PLOS Biology looks for in Methods and Resources articles: "Methods and Resources Articles describe technical innovations, including novel approaches to a previously inaccessible biological innovation, or substantial improvements over previously established methods. The reported method should be thoroughly validated, and while presenting new biological insights is encouraged, this is not a requirement for consideration."

Please see here (https://journals.plos.org/plosbiology/s/what-we-publish) for further information about PLOS Biology's article types.

Given the extent of revision needed, we cannot make a decision about publication until we have seen the revised manuscript and your response to the reviewers' comments. Your revised manuscript is likely to be sent for further evaluation by all or a subset of the reviewers.

**IMPORTANT - SUBMITTING YOUR REVISION**

*Re-submission Checklist*

*Published Peer Review*

*PLOS Data Policy*

*Blot and Gel Data Policy*

Sincerely,

Taylor

Taylor Hart, PhD,

Associate Editor

PLOS Biology

thart@plos.org

REVIEWS:

Reviewer #1: In this manuscript, the authors presented their meta-analysis on publicly available primary human developing brain scRNA-seq data to explore age-related features, and to establish age predictors given tissue composition or pseudobulk transcriptomic profiles. The authors also showcased how the obtained cell type-asgnostic age predictor can be applied to human neural organoids and mouse brain data to reveal developmental progression. This is an interesting exploration and provided a summary model that is applicable to other studies. On the other hand, the biological insight presented in the current manuscript are not really outstanding and novel. Some interpretations of the models are questionable and require additional clarification or validations. In addition, the method used to establish the models are not very innovative. The use case examples are valid and provide useful summarized information, but can't provide detailed and mechanistic insight, and thus represent only limited applications. Considering the mentioned limitations of the current manuscript, I'm afriad that I couldn't be more positive on the manuscipt, at least for its current form. In my opinion, tt needs great improvement, either on the biological insight or on the technical innovation, to be able to get accepted in PLOS Biology.

Followings are my detailed comments.

1. The tissue-level analysis is generally solid. The results are not surprising, that study-specific differences impact generalization. I would expect more about searching for and discussion about the technical or biological factors that affect such generalization. Is it mostly due to regional differences? Or is it mainly the technical variations, such as the dissection procedures that different studies may have dissected different parts of the tissues which have different enrichment of cell types? This is also relevant to the finding by the authors that astrocyte and progenitor proportions can predict well the developmental age, as astrocytes migration during development are reported to be more erratic (Tabata, Nat Comm. 2022). Also, as the analysis was based on the author-provided metadata, the authors should firstly verify whether the cell type annotations by different authors are compariable without needing additional effort to harmonize.

2. The regularized regression model used to establish the age prediction models are powerful and commonly used, but it is an old school method without substantial technical innovation, and it has been used before for relevant topics, e.g. neuronal maturation (He, BMC Genomics 2018). Of course, the algorithm behind the scenes doesn't need to be new and fancy, but without substantial technical innovation, in-depth biological insights should be presented. Although the authors did try to look into the established model and interpert its biological implication, the reported results are not very exciting either, and no biological followup was done or proposed.

3. The finding that the cell type-agnostic model performs better than the cell-type specific models is an interesting finding. The authors proposed that it was because the cell type-agnostic model, by including data of different cell types, captured different age-related genetic programs. However, another technical explanation is that the cell type-agnostic model was trained using way more data than the cell type-specific models. The authors should firstly show that the cell type-agnostic model performs better with the training dataset size controlled.

4. The genes selected by the cell type-agnostic model are confusing. Many of genes reported in Fig.4A with the highest importance, such as OAZ3, VMO1, AMIGO2 and CEL, don't show high expression in any of the developing brain cell types in both mouse (La Manno, Nature 2021) and human (Braun, Science 2023). This makes me wonder how much biological insight one can actually get by looking into those genes, and how reliable they are.

5. The authors looked into genes selected by the cell type-agnostic model for the enriched biological functions, which I appreciate. Meanwhile I think more should be done by looking into differennces and similarities of those biological processes among different cell types, and also to compare with the genes selected by different cell type-specific models. In my opinion this is very crucial, considering that the enriched functions are predominately neuron related, while it shows good performance also in the non-neuronal cell types. In fact, this concern, together with my last two comments, make me wonder whether there is a possibility that the cell type-agnostic model actually captures the ambient RNA background which is constance across cell types, likely different from one sample to another with the average transcriptomic changes across ages embedded, and probably neuron-biased (as neurons are more fragile and more likely to be damaged during tissue dissociation).

6. The authors showed user cases of applying the cell type-agnostic model to human neural organoids and mouse data, which provided reasonable results. My concern here, though, is whether the model actually provides more insight than simplier metrics, such as summarized module scores of maturation related pathways and gene modules (e.g. neuron projection, neurontransmitter secretion). Also, while the predicted ages may act as a useful summary metric, it doesn't seem to help on understanding any mechanism or providing in-depth biological insight.

Reviewer #2: Summary

The authors aimed to develop a general purpose metric to describe neurodevelopmental stages or developmental age across diverse single-cell datasets. They analyzed 13 single-cell RNA-seq datasets from the developing human brain, across 151 donors and 2.8M cells. Their analysis compares the relative contirbutions of tissue cell type composition and transcriptomic maturation in predicting developmental age.

They show that developmental changes in cell type composition can be predictive of developmental age within individual datasets but not across datasets. Since principal components derived from gene expression are also primarily a function of cell type composition, they tested PCs and found that they were not informative for this task. Next, the authors focused instead on specific cell type populations and found that astrocytes and progenitor cells consistently predicted neurodevelopmental age across dataset. In addition, they train a cell type agnostic model to learn a general signature of neurodevelopment. The cell type agnostic model performed better than models trained on individual cell types when evaluating across datasets. It weighed 462 genes as robustly able to predict developmental age, many of which were not strongly correlated with age individually. Lastly, the authors applied their cell type agnostic models to predict developmental progression in human neural organoids and embryonic mouse brain development.

Comments

- This paper makes important claims about the capabilities of different models, but it lacks a clear description of what level of performance is meaningful for downstream tasks and evaluation. How much error is acceptable? For instance, what is the difference between 1 week and 6 weeks? In lines 94-95, the authors report a median error of 1 week as an indicator of good performance, while in lines 125-126, a median error of 4 weeks is presented as good performance when training on astrocyte proportion. While I understand that this is relative to performance across cell types, it remains unclear what threshold of accuracy would make the results useful for biological applications.

- The significant difference between the cell type agnostic and cell type specific models seems to indicate that there is much more benefit that these models can obtain from training over larger datasets and being less biased in feature selection. However, training and evaluating is only performed on subsets of 13 datasets and thus does not seem fully conclusive of the claims being made and also whether performance could be significantly improved further. Without such a benchmark, it is difficult to assess whether the reported performance is truly robust across a wide range of datasets.

- Why was the cell type-agnostic model not trained on the full set of protein-coding genes? Did the authors test more complex architectures (e.g. non-linear models) for handling a feature space of over 10,000 genes? Did they try deep learning methods that do not require any feature selection? Furthermore, how did they address cross-species transfer when shared genes were absent?

- Given the different models tested, which core model do the authors recommend using? It would be helpful to directly compare the cell type-agnostic model with the models trained on tissue composition. Some evaluations use MAE (in weeks), while others use Pearson R. Consistency in evaluation metrics would strengthen the paper. Could the authors create a final plot comparing all approaches tested for predicting neurodevelopmental age? Additionally, are there existing baseline models for predicting age that could be included for comparison? If the cell type-agnostic model performs best, what advantage does the predictor based on cell type composition provide? Is there a clear benefit of using one over the other?

- In the analysis identifying astrocyte proportions as predictive of developmental age, did the authors perform multiple hypothesis test correction (Fig. 2D, 2F)? The performance improvement does not appear very significant and could reflect multiple testing across many cell types.

Reviewer #3: This manuscript develops several statistical models to predict single-cell and tissue transcriptomic ages during early development. The predicted ages can serve as a clock to assess developmental timings across different studies, which is of significant interest to the community. The authors integrated multiple single-cell RNA-seq datasets, including 2.8 million cells, and validated their predictions using different datasets. Notably, they created a cell-type-agnostic model that does not require cell type information to estimate transcriptomic age for entire single-cell datasets. The manuscript also applies these models to organoid data and further examines mouse embryonic development.

Overall, the study was logically designed from tissue to single-cell modeling. I have several comments on the methodological rigor, biological significance, and usage.

1. It is unclear what glmnet regularization(s) the cell-type-agnostic model uses.

2. The cell-type-specific models are insufficiently described. How many genes did they input, all 10k genes like agnostic model or cell type marker genes? It is a surprise that Fig 3 shows a lack of cell type specificity if cell type marker genes were used.

3. Also, did the cell-type spec model use glmnet, the same regularized model as the agnostic one?

4. How conserved (or spec) are co-expression modules of 462 cell-type agnostic genes across different. studies? Is it also possible to calculate/assess the modular ages beyond individual genes/cells, which may provide additional insights into functional/mechanistic aging?

5. Moreover, how do those 462 genes relate to tissue PCs if they are free to cell types?

6. Is it possible to comment on how a cell-type agnostic model may help identify novel cell types and aging?

7. The prediction for organoids has relatively large ranges (weeks of variation, Fig 7)

8. Some performance comparisons need further justifications for rigor, such as 30.6% astrocytes for GO term models, R=-0.38 w/o significance, 3-week cutoff for AUROC in Fig 7, etc.

9. In the tissue modeling, it is unclear how sensitive cell class/type annotation can affect the age prediction via cell compositions. Additionally, the tissue-regularized regression models should be elaborated and compared with cell-type models.

Reviewer #4: In this manuscript, Venkatesan et al. present a large-scale meta-analysis of 13 human fetal brain scRNA-seq datasets (~2.9 million cells) aimed at disentangling tissue-level composition effects from cell-autonomous maturation and at developing broadly generalizable developmental age predictors. The authors show that study-specific composition and PC-based age proxies perform well within individual datasets but fail to generalize across studies, largely because PCs capture shifts in cell type proportions. To address this, they derive a remarkably simple and transferable compositional model based on the relative increase of astrocytes and decrease of progenitors, and complement it with a regularized, cell type-agnostic transcriptomic model that reliably predicts age across cell types, datasets, organoids, and even mouse embryonic brain. Together with the freely available code, these models provide a practical framework for aligning neurodevelopmental tempo across systems.

1) A key question is how many datasets are minimally required to achieve robust performance. A subsampling analysis using the existing 13 datasets would be informative, for instance by training region-specific models (e.g. a cortex-dominant model) and comparing their performance against the full model on held-out non-cortical datasets, such as the hypothalamus data in Figure 3E. This would directly test whether the model's generalizability depends on the anatomical diversity of the training set.

2) Several of the reported differences appear to be within the variance or error range of the analysis approach. All such comparisons should be supported with appropriate statistics, including effect sizes and p-values. Since the models generally have an error of a few weeks, it would also be helpful for the authors to discuss under which circumstances this level of accuracy is sufficient and in which contexts it may be too coarse.

3) The authors note that organoids tend to display greater variability than primary tissues. It would strengthen the study to quantify this explicitly at the level of developmental age predictions.

4) Many of the models plateau at older developmental ages (e.g., Fig. 1E, Fig. 3D+E, Supplementary Fig. 7A). For organoids (Supplementary Fig. 7B), the authors attribute this to metabolic constraints, yet similar flattening is apparent in the fetal brain data. Additional analyses or discussion should clarify whether this reflects biological features, limited sample availability, or computational artefacts.

5) The manuscript would benefit from a more comprehensive analysis of prediction errors across developmental time. Reporting only median errors masks potential biases. Stratifying the dataset into early, mid, and late developmental stages and recalculating prediction errors for each would reveal whether accuracy is uniform across development or biased toward particular stages.

6) The elegant linear mapping between mouse embryonic days and human gestational weeks calls for sensitivity analyses. Testing alternative ortholog sets and excluding lineage-biased gene modules would help determine whether cross-species tempo estimates are robust or disproportionately driven by specific gene groups.

7) The finding of a tenfold faster developmental tempo in mice compared to humans appears strikingly high, especially relative to previous estimates of about two- to fivefold acceleration (e.g. Rayon et al. 2020, Iwata et al. 2023). Additional evidence supporting this larger factor should be cited or the claim more cautiously framed.

8) To test the limits of generalizability, it would be valuable to develop a parallel cell type-agnostic model trained exclusively on mouse brain development and to compare its predictions with those of the human-trained model on the same mouse datasets. This would provide an internal benchmark for cross-species applicability and could refine the estimates of relative tempo between species.

9) On page 4, line 137, the authors mention two coefficients. These should be directly referenced to the corresponding tables or figures where the values are presented to aid clarity.

10) Finally, while fully optional, a direct experimental test in human neural organoids would be highly compelling. For example, subjecting organoids at a fixed developmental stage to a defined stressor such as hypoxia and then measuring the predicted age shift relative to controls would provide strong evidence that the model captures state-dependent changes independent of chronological time.

---

## [Decision Letter · Decision Letter 2]

16 Mar 2026

Dear Dr Gillis,

Thank you for your patience while we considered your revised manuscript "Cell Type-Agnostic Transcriptomic Signatures Enable Uniform Comparisons of Neurodevelopment" for consideration as a Methods and Resources article at PLOS Biology. Your revised study has now been evaluated by the PLOS Biology editors, the Academic Editor, and the original reviewers.

In light of the reviews, which you will find at the end of this email, we are pleased to offer you the opportunity to address the remaining points from the reviewers in a revision that we anticipate should not take you very long. We will then assess your revised manuscript and your response to the reviewers' comments with our Academic Editor aiming to avoid further rounds of peer-review, although we might need to consult with the reviewers, depending on the nature of the revisions.

**IMPORTANT - SUBMITTING YOUR REVISION**

*Resubmission Checklist*

*Published Peer Review*

*PLOS Data Policy*

*Blot and Gel Data Policy*

Sincerely,

Taylor

Taylor Hart, PhD,

Associate Editor

PLOS Biology

thart@plos.org

REVIEWS:

Reviewer #1: I very much appreciate the effort by the authors to revise the manuscript and address the concerns from me as well as other reviewers. I think the manuscript has been improved, especially from the biological perspective, that top genes from the cell type-agnostic model now show much more reasonable functions and expression patterns.

I still have a relatively minor concern on this topic though. As the authors reported, the reason why the lowly expressed genes were reported earlier was because coefficients are scaled inversely with gene expression. I assume the authors didn't retrain the model using a different approach in this revised manuscript, just to use a different way to assess feature importance. Since the author used elastic net for regularized regression, and what elastic net regularizes on is the coefficients, does it mean that the unintentionally high coefficients from the lowly expressed genes squeeze the contribution of other genes in the model? Would the model be further improved if the lowly expressed genes are filtered beforehand, or instead of using gene expression as the input, using principal components as a denoised input? For the latter, the PCA loadings can be saved to apply the same transformation to the queue data for model application. This may also be helpful considering the noisy nature of scRNA-seq data.

Reviewer #2: The authors have meaningfully addressed my comments

Reviewer #3: The authors addressed almost all of my previous comments. Regarding the large variation in predicted ages in organoids, Fig. 7 is still unclear about how within-cell-type (or within-sample?) variability compares with the fetal variability. Also, the calculation of this dispersion of predicted developmental age (SD) conditioned on observed age should be elaborated, especially about "conditioned on".

Reviewer #4: In their revised manuscript, the authors have appropriately addressed all reviewers' comments.

---

## [Editor Report · Decision Letter 3]

24 Mar 2026

Dear Dr Gillis,

Thank you for your patience while we considered your revised manuscript "Cell Type-Agnostic Transcriptomic Signatures Enable Uniform Comparisons of Neurodevelopment" for publication as a Methods and Resources at PLOS Biology. This revised version of your manuscript has been evaluated by the PLOS Biology editors and the Academic Editor.

Based on our Academic Editor's assessment of your revision, we are likely to accept this manuscript for publication. Please also make sure to address the following data and other policy-related requests.

IMPORTANT: Please ensure that your next revision addresses the following points:

**Title:

We are not opposed to your paper's current title, but we propose an alternative formulation for your consideration: we wonder if the title would be improved by substituting the word "neurodevelopment" with "neural maturation"? If you opt to change the title, please change it in the form and in the manuscript document when you submit your revision.

**Data & Code:

Thank you for including data and code on Github. We require a few more things to meet our requirements for data and code provision.

First, we require the numerical values as they underly the figure plots in the paper. Please include (either in your online repository, or as a supplementary file S1 Data / "S1_Data.xlsx") the values corresponding to the following figure plots:

2DF

3C(dot plot)F

4C

7ACD

S6ABC

S7

S8

S11(right side)

S12A

S13

S15A(right side)B(right side)

S16

Because Github depositions can be readily changed or deleted, we require that code be made available via a permanent DOI’d copy (e.g. in Zenodo) and that this URL be provided in the manuscript and Data Availability Statement, instead of the Github URL. Please also choose a license for your code.

Please finalize the Data Statement for publication.

**Supplement:

Please only upload one copy of each figure.

Please adopt our standard naming scheme for the supplementary tables, eg name them "S1 Table"; "S2 Table", etc. with the filenames according to this format "S1_Table.xlsx", "S2_Table.csv", etc.

We expect to receive your revised manuscript within two weeks.

*Published Peer Review History*

*Press*

Sincerely,

Taylor

Taylor Hart, PhD,

Associate Editor

thart@plos.org

PLOS Biology

---

## [Editor Report · Decision Letter 4]

31 Mar 2026

Dear Dr Gillis,

Thank you for the submission of your revised Methods and Resources "Cell Type-Agnostic Transcriptomic Signatures Enable Uniform Comparisons of Neural Maturation" for publication in PLOS Biology. On behalf of my colleagues and the Academic Editor, Selene Fernández Valverde, I am pleased to say that we can in principle accept your manuscript for publication, provided you address any remaining formatting and reporting issues. These will be detailed in an email you should receive within 2-3 business days from our colleagues in the journal operations team; no action is required from you until then. Please note that we will not be able to formally accept your manuscript and schedule it for publication until you have completed any requested changes.

PRESS

Sincerely,

Taylor

Taylor Hart, PhD,

Associate Editor

PLOS Biology

thart@plos.org